# Deep photonic network platform enabling arbitrary and broadband optical functionality

Ali Najjar Amiri[1], Aycan Deniz Vit [1], Kazim Gorgulu[1] & Emir Salih Magden [1]✉

Expanding applications in optical communications, computing, and sensing continue to drive the need for high-performance integrated photonic components. Designing these on-chip systems with arbitrary functionality requires beyond what is possible with physical intuition, for which machine learning-based methods have recently become popular. However, computational demands for physically accurate device simulations present critical challenges, significantly limiting scalability and design flexibility of these methods. Here, we present a highly-scalable, physics-informed design platform for on-chip optical systems with arbitrary functionality, based on deep photonic networks of custom-designed Mach-Zehnder interferometers. Leveraging this platform, we demonstrate ultra-broadband power splitters and a spectral duplexer, each designed within two minutes. The devices exhibit state-of-the-art experimental performance with insertion losses below 0.66 dB, and 1-dB bandwidths exceeding 120 nm. This platform provides a tractable path towards systematic, large-scale photonic system design, enabling custom power, phase, and dispersion profiles for high-throughput communications, quantum information processing, and medical/biological sensing applications.

Photonic integrated circuits (PICs)[1,2] have significantly evolved over the last decade and are now essential technological components with critical importance in optical communications[3], sensing[4,5], and computing[6–8]. With the growing diversity and complexity of photonic applications, designing custom PICs with state-of-the-art performance metrics has become one of the most critical drivers of advancement in photonic systems. Traditional approaches relying on prior knowledge of relevant architectures, fundamental principles, and physical intuition yield a limited library of known devices and severely restrict the potential capabilities of the resulting photonic systems. More general approaches have recently emerged under the broad category of inverse/machine-optimized design[9–14], allowing for greater design flexibility than manual tuning of waveguide parameters. Through comprehensive searches over the complete domain of fabrication-compatible devices, various types of couplers[15], polarization splitters[10,16], and spectral filters[9,14] have been proposed and demonstrated through these inverse-design methods. However, in these

"free-form" design approaches, the degrees of design freedom are effectively controlled by the specified device footprint, which has key implications on the final device performance and the associated computational cost. While larger device footprints inherently provide the necessary design flexibility for complex and arbitrary optical functionality, they also rapidly scale the computational complexity of the necessary optimization process due to the physically-accurate electromagnetic simulations required[9,10,14,17]. These requirements preclude the design of arbitrarily complex, ultra-broadband, or wavelength-specific photonic devices for the increasing number and variety of use cases and application requirements.

The ideal approach to photonic design must allow for arbitrarily-specified photonic functionality while maintaining low computational cost. In recent years, programmable PICs made from Mach-Zehnder interferometers (MZIs) have been proposed as a potential solution to this problem[3,18–21]. These systems enable tuning of optical responses through active phase shifters to achieve wavelength-specific linear

[1]Department of Electrical and Electronics Engineering, Koç University, Sariyer, Istanbul 34450, Turkey. ✉e-mail: esmagden@ku.edu.tr

mappings for applications including high-speed and power-efficient optical signal routing[3,20,22], image/signal classification[23–27], and quantum computing[8,28]. Yet, the potential utility of photonic interferometer networks extends well beyond these demonstrated capabilities, with critical implications towards the design of photonic systems with arbitrarily complex transfer functions.

In this paper, we introduce and experimentally demonstrate a highly-scalable framework for the design of photonic systems with arbitrarily-specified functionality, based on a deep photonic network architecture of custom-designed MZIs. Our architecture consists of a mesh of individually designed interferometers and is modeled by an equivalent computational network equipped with ultra-fast and physically-accurate simulation capabilities. In this network, each MZI is constructed from unique waveguide tapers, allowing for specific wavelength-dependent phase profiles to be achieved according to the target photonic functionality specified. The exact geometry of the individual interferometers is optimized by leveraging physics-informed machine learning capabilities in our design framework through a combination of rapid lookup of waveguide parameters and successive evaluation of photonic transfer matrices. Using this framework, we design ultra-broadband 50/50 and 75/25 power splitters and a spectral combiner/splitter, each in less than two minutes, with inherent fabrication compatibility on the 220-nm-thick silicon-on-insulator platform, and experimentally demonstrate state-of-the-art performance for all three devices. Our presented framework provides a path towards the systematic design of large-scale photonic systems with arbitrarily-specified, wavelength-dependent, or ultra-broadband responses.

## Results

### Deep photonic network architecture

The architecture of our deep photonic network consists of an input layer, a series of MZI layers, and an output layer, as shown in the schematic in Fig. 1a. This architecture based on a mesh of MZIs has the theoretical capability to implement any linear $N \times N$ input-output mapping in order to achieve arbitrary optical functionality[29–31]. Input optical signal to the network is provided either externally by a series of

couplers as shown, or by waveguides from upstream devices on-chip. The input optical signal is processed unidirectionally through layers of custom MZI interferometers, each with its own specific $2 \times 2$ mapping function denoted by $T_{i,j}$. This modular network is modeled using the transfer matrix description of each one of its constituent building blocks, in a modular configuration. Specifically, each MZI consists of two pairs of waveguide tapers with custom geometries and two directional couplers, as illustrated in Fig. 1(b). The overall transfer matrix for each MZI is described by the transfer matrices of these constituent blocks as:

$$
\begin{aligned}
T(\lambda) = e^{-j\varphi(\lambda)} &\begin{bmatrix} t(\lambda) & -jq(\lambda) \\ -jq(\lambda) & t(\lambda) \end{bmatrix} \begin{bmatrix} e^{-j\theta_{21}(\lambda)} & 0 \\ 0 & e^{-j\theta_{22}(\lambda)} \end{bmatrix} \\
\times\, e^{-j\varphi(\lambda)} &\begin{bmatrix} t(\lambda) & -jq(\lambda) \\ -jq(\lambda) & t(\lambda) \end{bmatrix} \begin{bmatrix} e^{-j\theta_{11}(\lambda)} & 0 \\ 0 & e^{-j\theta_{12}(\lambda)} \end{bmatrix}
\end{aligned}
\tag{1}
$$

where $t(\lambda)$, $q(\lambda)$, and $\varphi(\lambda)$ are the through- and cross-port amplitude coefficients and the phase response of the directional couplers, and $\theta_{11}(\lambda)$ through $\theta_{22}(\lambda)$ are the phases accumulated in corresponding waveguide tapers. The wavelength dependence of each one of these parameters plays a critical role in achieving arbitrary optical functionality in our networks. The directional couplers used throughout the network are identical and are designed to be approximately 50% couplers at 1550 nm (see Supplementary Section 1 for details). A schematic of this directional coupler and its simulated through-port transmission are shown in Fig. 1c. In contrast, all waveguide tapers are unique and custom-designed using a set of width and length parameters, as illustrated in Fig. 1d, which are determined through an iterative optimization algorithm. The phase accumulated through each custom waveguide taper is calculated as a differentiable function of these custom widths ($w_i$), taper length ($L_\theta$), and input wavelength ($\lambda$), using the waveguide effective index $n_{\text{eff}}(w, \lambda)$. This unique implementation allows the network to achieve wavelength-dependent phase profiles different from that of a straight waveguide, as demonstrated in the inset of Fig. 1d, enabling much higher degrees of freedom while maintaining the same device footprint. Our design

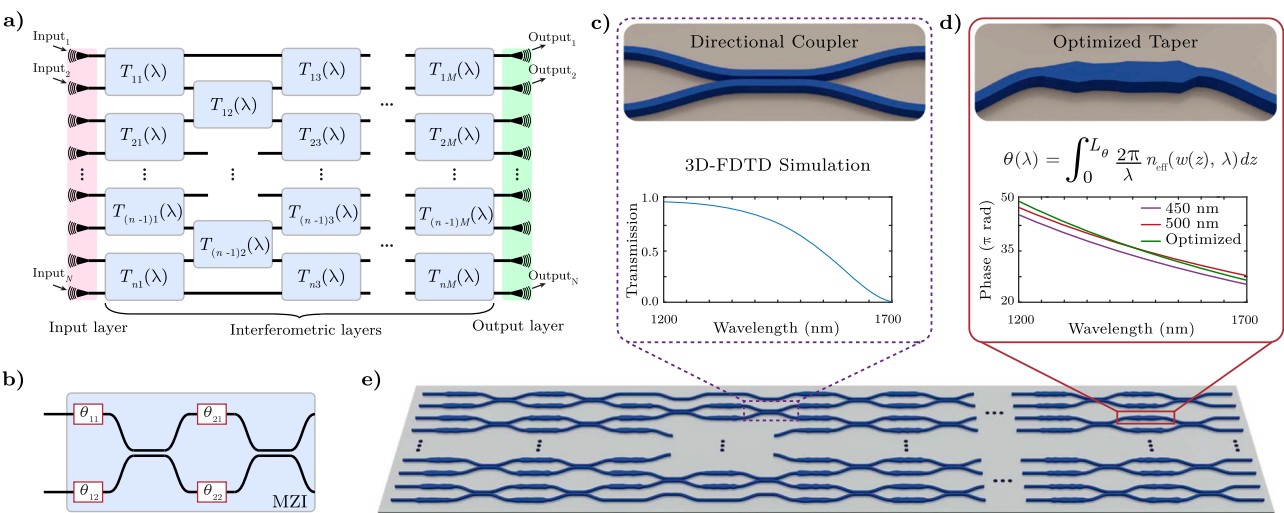

**Fig. 1 | Deep photonic network architecture and components. a** The network architecture is composed of the input stage, horizontally-cascaded and vertically-repeated custom interferometric layers, and the output couplers. Each interferometric layer consists of a combination of Mach-Zehnder interferometers and individually-optimized waveguide structures. **b** Block diagram of a Mach-Zehnder interferometer with two pairs of waveguide tapers of custom geometries and two directional couplers. $\theta_{11}$ through $\theta_{22}$ indicate the phases accumulated through each custom waveguide taper. **c** Schematic of the directional coupler with two S-bends

and a 10 μm-long coupling section, and its 3D-FDTD simulated transmission response. **d** Schematic of an example custom waveguide taper constructed from a set of optimizable width parameters, from which the accumulated phase is calculated as a function of wavelength using the effective index. These custom waveguide tapers enable unique spectral phase profiles different from those in straight waveguides, as shown in the inset. **e** Overall structure of an example deep photonic network with cascaded interferometric layers of directional couplers and individually optimized waveguide tapers.

framework then constructs the overall photonic integrated circuit through an arbitrary number of interferometric layers as shown in Fig. 1e.

## Simulation and optimization of the network's optical response

Propagation of the complex optical amplitude through the network is carried out by a computational graph mimicking the physical network architecture. At each wavelength, the optical transformation carried out by the mesh of interferometers between $N$ input channels and $N$ output channels is represented by a computational graph. This architecture calculates the wavelength-dependent linear scattering matrix $S(\lambda)$ of the entire deep photonic network according to

$$
\prod_{q=1}^{\lceil \frac{M}{2} \rceil}
\begin{bmatrix}
T_{1,R+1-2q} & & & \\
& \ddots & & \\
& & & \\
& & & T_{n,R+1-2q}
\end{bmatrix}
\times
\begin{bmatrix}
F & & & \\
& T_{1,R-2q} & & \\
& & \ddots & \\
& & & T_{n-1,R-2q} \\
& & & & F
\end{bmatrix}
\tag{2}
$$

for networks with an even number of inputs, and by

$$
\prod_{q=1}^{\lceil \frac{M}{2} \rceil}
\begin{bmatrix}
F & & & \\
& T_{1,R+1-2q} & & \\
& & \ddots & \\
& & & T_{n,R+1-2q}
\end{bmatrix}
\times
\begin{bmatrix}
T_{1,R-2q} & & & \\
& \ddots & & \\
& & T_{n,R-2q} & \\
& & & F
\end{bmatrix}
\tag{3}
$$

for networks with an odd number of inputs. Here, $M$ represents the number of interferometric layers, $n = \lfloor \frac{N}{2} \rfloor$, $R = 2\lceil \frac{M}{2} \rceil + 1$, and $F$ is a scalar indicating the phase accumulated through the topmost and bottom-most arms of the network where no interferometer is present. Note that the very first matrix is omitted from the products when using an odd number of layers.

This computation involves integrating the waveguide effective index using the custom widths and lengths for each waveguide taper, and extracting the directional coupler through-port, cross-port coefficients, and phase response from the 3D-FDTD results. In order for our custom photonic networks to be optimized for user-defined optical functionality, these operations are implemented through a differentiable programming construct, enabling both fast parameter lookups and automatic calculation of relevant derivatives[32]. For calculating $\theta(\lambda)$, we numerically integrate the effective index throughout the length of the custom tapers using data obtained from Silicon Photonics Toolkit[33], an open-source software package providing access to several important propagation-related parameters in silicon waveguides as functions of wavelength and waveguide width. The directional coupler coefficients are similarly extracted from a differentiable interpolation of its 3D-FDTD simulation results. The result of this computation yields the complete network transfer function with a high degree of physical accuracy including the wavelength-dependent mappings for each input-output pair.

The ability to rapidly calculate a given network's optical response as a differentiable function of its design parameters is critical from an optimization perspective. Using this capability, we construct an optimization procedure by iteratively modifying the waveguide tapers in order to obtain application-specific photonic networks with arbitrarily defined transfer functions. This procedure is illustrated for an example 1-input 4-output network in Fig. 2a. First, we initialize a network with the desired number of interferometric layers and input-output ports. We define the target optical transfer function of these input-output pairs ($T_{\text{target}}(\lambda)$), and assign semi-random width and length parameters to the constituent custom waveguide tapers. The network's optical response is evaluated as a function of wavelength using the procedure described above and compared with the target transfer function. The difference between the calculated and target transfer functions is formulated as a mean squared error $J(x) = \frac{1}{Q}\sum_\lambda |T_{\text{calculated}}(\lambda,x) - T_{\text{target}}(\lambda)|^2$, where $Q$ is the number of wavelengths and $x$ are design parameters including widths and lengths of the custom tapers. Gradient of $J(x)$ with respect to these design parameters $\nabla_x J$ is calculated through a back-propagation procedure. We then minimize this error by iteratively modifying the widths and lengths of waveguide tapers, as illustrated in Fig. 2b, using a gradient-based optimization algorithm[34]. In addition to this error itself, we implemented numerous regularization schemes to achieve inherent fabrication compatibility by restricting waveguide widths from undergoing extreme changes in the custom tapers throughout the optimization procedure. Details regarding network initialization, convergence of this optimization process, and final resulting waveguide parameters can be found in Supplementary Section 2.

## Arbitrary optical functionality with deep photonic networks

One of the key advantages of our proposed deep photonic network functionality is its ability to enable designs of photonic devices with arbitrary spectral specifications. We demonstrate how this capability allows for a universal design procedure for designing devices with ultra-broadband responses, and also devices with specific spectral features. As a proof of principle, this functionality is illustrated in Fig. 3 with three separate devices: two broadband power splitters with 50/50 and 75/25 splitting ratios operating within 1400-1600 nm, and a $1 \times 2$ spectral duplexer between 1450 nm and 1630 nm.

Depending on the complexity of the desired functionality, our framework allows for the appropriate selection of hyperparameters of the deep photonic network including the number of interferometric layers and the number of custom widths in each waveguide taper. Details regarding the selection of hyperparameters can be found in Supplementary Section 3. Here, the power splitters are both designed with networks of three layers each, and the duplexer is designed with a network of six layers. For each custom waveguide taper in our devices, we used five trainable widths and a trainable length, resulting in a total of 24 parameters for each MZI in our photonic networks. The evolution of the resulting mean squared errors throughout the optimization processes are plotted in Fig. 3a–c, where convergence is achieved in several hundred iterations and, at most, a few minutes on a single Tesla V100 GPU. Details regarding the optimization time of the photonic networks and their scalability can be found in Supplementary Section 4.

The wavelength-dependent design capability of our network is illustrated in Fig. 3d–f, where we plot the transmission at one of the output ports for each one of the three devices as a function of wavelength. Throughout optimization, the output state evolves towards the target output functionality, as can be seen by the optical responses gradually approaching the desired 50%, 25% (for one output), and the spectrally duplexed outputs for the three devices, respectively. In Fig. 3g–i, the transmission spectra at both output ports are plotted for each device at their randomly-initialized states at the beginning of optimization, at an intermediate state where the devices have been partially trained, and at the final states of the optimized devices. The

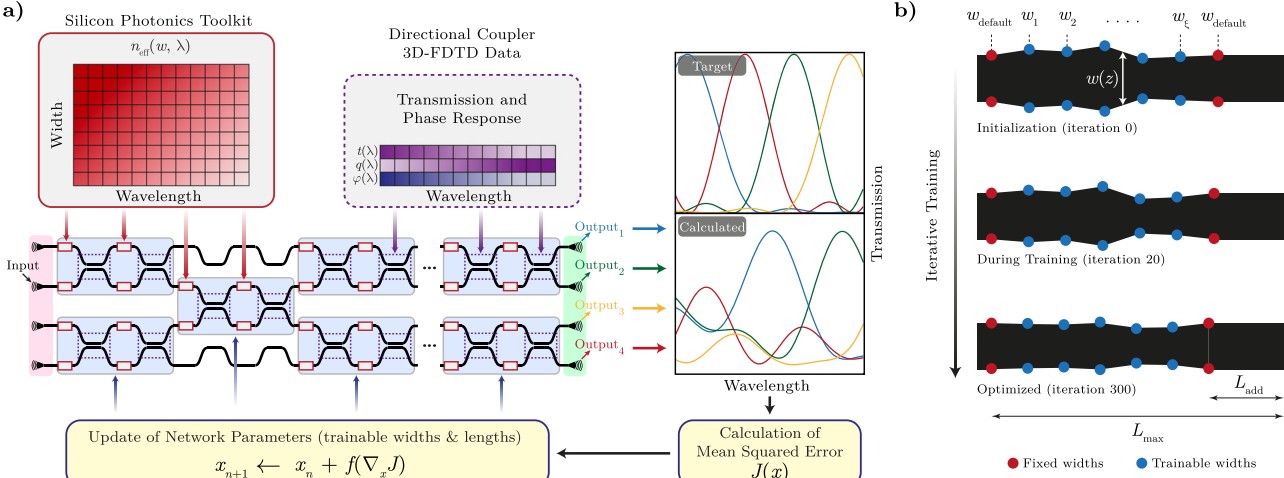

**Fig. 2 | Optimization of an example 1-input 4-output photonic network. a** The 1 × 4 network structure is created with the desired number of layers, randomly-intialized custom waveguide tapers (red rectangles), and a target transmission response for input-output pairs. The mean squared error is computed from the difference between the calculated and target transfer functions, by summing over the specified wavelength range. The network parameters are trained iteratively through a backpropagation algorithm using the gradient of this error with respect to the design parameters denoted by $x$ in the custom waveguide tapers. Other network components including the directional couplers and input/output layers are not trainable. **b** Evolution of a custom waveguide taper throughout optimization of the deep photonic network, where its geometry is shown at random initialization, at iteration 20, and at the end of optimization. Fixed widths ($w_{default}$) and trainable widths ($w_1, w_2, ..., w_\xi$) are marked with red and blue circles along the taper, respectively. At each iteration, an additional straight waveguide of length $L_{add}$ is inserted at the end of the custom taper in order to achieve matching $L_{max} = 10\ \mu m$ lengths for all tapers.

final device responses demonstrate a near-perfect match with the specified target functionality. These responses are verified by the propagation of the optical input in the final optimized devices, which are plotted using the electric field intensity from 3D-FDTD simulations in Fig. 3j–l. These simulation results confirm the expected outputs from our transfer matrix calculations that our networks are trained with. As expected, the power splitters achieve broadband operation; and the duplexer functions as a spectral splitter within its spectral design range, providing long-pass and short-pass outputs.

**Experimental demonstration and analysis of network response**
The experimental characterization results for the two power splitters are shown in Fig. 4a, b. For the 50/50 splitter, the maximum deviation from 50% transmission is as low as ± 6.42% for both output ports; and the insertion loss is measured to be less than 0.5 dB. As such, our network-based power splitter experimentally achieves a deviation of at most 0.6 dB within the 120 nm of measured bandwidth, and therefore a 1-dB bandwidth much wider than that. Similarly, for the 75/25 splitter, the deviations from the target transmission are within ±5.49% ( ±0.86 dB) and ±8.88% ( ±0.55 dB), for output ports number one and number two, respectively. The measured insertion loss is less than 0.61 dB for both output ports. These results indicate that the 75/25 splitter achieves a 1-dB bandwidth of at least 120 nm, our widest measurement range possible. The spectral duplexer's experimental characterization results are shown in Fig. 4c. Within the pass-bands, a maximum loss of 11.45% (0.52 dB) and 15.30% (0.72 dB) are measured for the short-pass and long-pass outputs, respectively, and the insertion loss is measured about 0.66 dB (occurring at 1590 nm). The measured cutoff wavelength is around 1555.2 nm, compared to the specified target cutoff wavelength of 1550 nm. The extinction ratio between the two outputs is better than 15 dB for the majority of the wavelength range characterized, and only reaches 13.6 dB at the edge of the measured spectrum (1600 nm). All three devices experimentally exhibit state-of-the-art performance and a close match with the training objective transmission responses. Reflection in our networks was also characterized, and found to be around -30 dB for the majority of the measured spectrum with no practical influence on our device optimization processes (Supplementary Section 5). These results

demonstrate and experimentally verify the universal capability of our design approach.

Next, we analyze the robustness of our deep photonic networks against fabrication variations. Specifically, we plot the resulting transmission responses from transfer matrix calculations under potential over-etch and under-etch scenarios in Fig. 4d–f up to a change of ±20 nm in the waveguide widths and gaps. The device responses are calculated by simulations of the network structures with updated waveguide tapers and directional couplers for the amounts of specified etch offsets. We observe minimal deviation of the transmission response from the ideal case with ± 10 nm over- and under-etch. At ±20 nm, we observe more significant changes in the simulated transmission responses, resulting from changes in the wavelength-dependent phase profiles in waveguide tapers and the shifted responses of the directional couplers, as expected. This is also demonstrated in Fig. 4g–i, where we plot the mean squared error of the resulting transmission with different over- and under-etch amounts. The calculated error increases with larger over-/under-etch amounts, indicating deteriorations in the resulting device performance. Functionally, we note that all three devices can still work as intended, with slightly inferior performance metrics up to the simulated ± 20 nm etch offsets.

**Deep photonic network capability and fabrication robustness**
The scalability of our deep photonic networks and the computational efficiency of our underlying simulation/optimization framework can provide highly capable networks with extremely large degrees of freedom to design arbitrarily complicated optical devices. For this architecture, the selection of the number of interferometric layers is a major design choice that determines the number of degrees of freedom for the network. While the trainability and capability of the resulting network increase with the number of layers at first, each additional layer also introduces additional propagation loss due to the waveguide bends added with each layer. This trade-off between device capability and insertion loss can be modeled by analyzing devices with different numbers of layers trained for the same objective functionality. In Fig. 5a, we plot the final mean squared error in the simulated transmission responses for different 50/50 power splitters designed

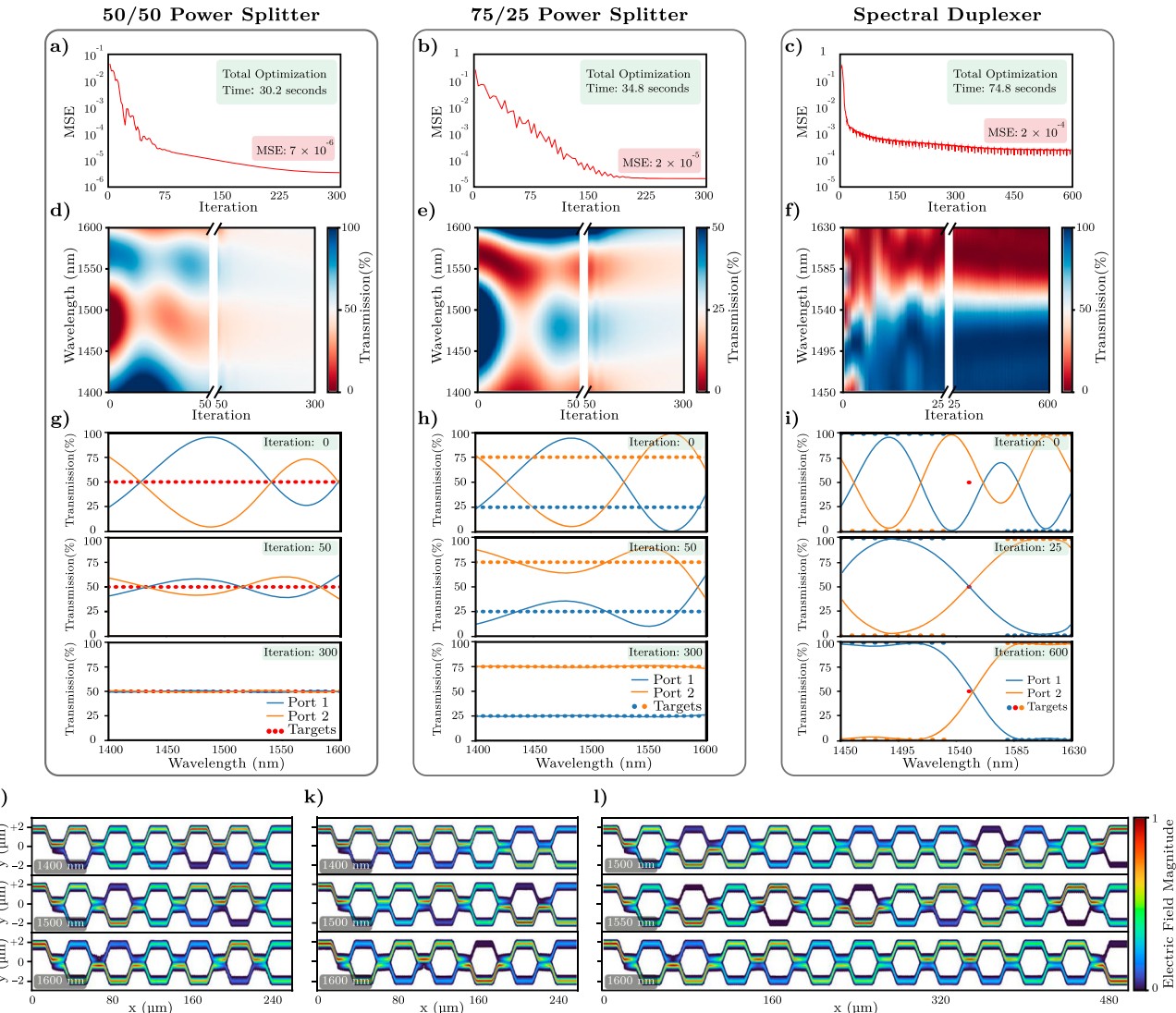

**Fig. 3 | Optimization and final simulation results of power splitter and spectral duplexer deep photonic networks.** The mean squared error (MSE) versus iteration throughout optimization of **a** a 50/50 power splitter with 3 layers of MZIs (72 trainable parameters, 240 μm device length), **b** a 75/25 power splitter with 3 layers of MZIs (72 trainable parameters, 240 μm device length), and **c** a spectral duplexer with 6 layers of MZIs (144 trainable parameters, 480 μm device length). All three devices converge in several hundred iterations, within 1-2 minutes. **d**–**f** Transmission at the designated output port of each device as a function of wavelength. The evolution of this transmission through the iterative training process enables all three devices to achieve near-perfect transfer functions by the end of optimization. **g**–**i** Transmission spectra for each output during optimizations show gradual convergence to the target transfer functions indicated by the circles. The power splitters are optimized with 32 evenly-spaced wavelengths between 1400-1600 nm, and the duplexer is optimized with 21 wavelengths between 1450-1630 nm with a target cutoff at 1550 nm. Magnitude of the electric field at three different wavelengths obtained from 3D-FDTD simulations confirming broadband and flat-top operation for **j** the 50/50 power splitter, **k** the 75/25 power splitter, and (**l**) the spectral duplexer.

with numbers of layers ranging from $M = 2$ to $M = 60$. As expected, the simulated error in the transmission response initially decreases and reaches a minimum with networks of 3 and 4 layers. However, with the increased number of layers, the accumulation of insertion loss through additional layers outweighs the benefits of increased network capability, and results in a larger calculated error and an inferior transmission response.

Similarly, while longer networks with more interferometric layers can provide larger degrees of freedom and more complex optical capabilities, they are also less robust to fabrication variations. Similar to the added optical loss, errors in the phase profiles add up through the additional layers and negatively affect the resulting device performance. We analyze the fabrication tolerance of 50/50 splitters constructed from different numbers of layers in Fig. 5b, where the mean squared error is plotted as a function of the etch offset. The results demonstrate that longer networks (with greater numbers of

layers) are more sensitive to fabrication-induced changes due to the accumulation of phase and coupling errors within consecutive MZIs. For instance, while the minimum error calculated is similar for 3-layer and 4-layer splitters, the 4-layer network exhibits significantly worse performance with etch offsets reaching ± 20 nm. This analysis serves as an important guideline towards determining the appropriate number of layers for the design of specific structures using the demonstrated custom networks. Similar analyses for the 75/25 power splitter and the spectral duplexer can be found in Supplementary Section 3.

**Multi-objective design capabilities**

In addition to creating a single device with a single optical functionality, our design framework is also capable of utilizing multi-objective capabilities to create devices with more complex optical functionalities. We use this particular approach to demonstrate the design of deep photonic networks with more advanced capabilities including

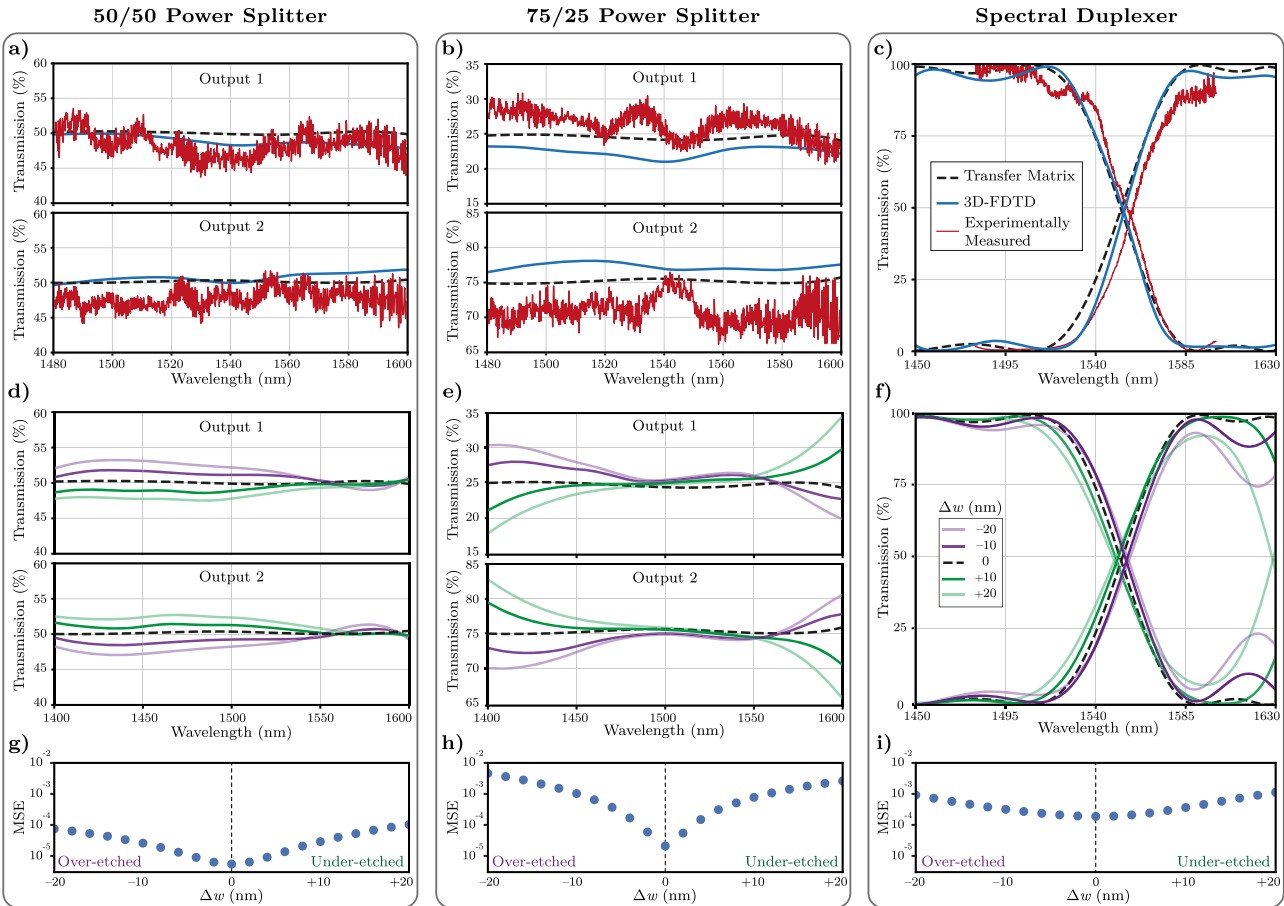

**Fig. 4 | Experimental measurements and fabrication tolerance analysis of deep photonic networks. a–c** Measured transmission results together with transfer matrices and 3D-FDTD simulations at the output ports of the power splitters and the spectral duplexer. All three devices demonstrate agreement with simulation results over wide bandwidths with flat-top and low-loss transmission responses. **d–f** Transfer-matrix analysis of robustness against fabrication-induced variations for 10 nm and 20 nm over-etch and under-etch cases for the three devices. All components including directional couplers, S-bends, and waveguide tapers, are uniformly modified in simulation with the indicated etch offsets. **g–i** Resulting mean squared error in devices subject to over-etch and under-etch variations. With ± 20 nm modification of the waveguide widths, the resulting error typically increases by 1-2 orders of magnitude, corresponding to the changes in the simulated transfer function of the respective devices.

built-in tolerance against fabrication variations as well as scalability through different optical transfer functions across a larger number input-output pairs. For achieving fabrication tolerance, we simultaneously optimize the optical response of multiple different versions of a network, each one resulting from a different over-etch or under-etch scenario. Moreover, we also configure a photonic network as a combination of multiple different power splitters, in which the optical response depends on what port the optical input is received at. In this case, we define a more general figure of merit as a mean squared error including all possible combinations of fabrication variations and input ports as $J(x) = \frac{1}{Q}\sum_{\Omega}\sum_{\Delta w}\sum_{\lambda}|T_{calculated}(\Omega,\Delta w,\lambda,x) - T_{target}(\Omega,\lambda)|^2$ where the width offset parameter $\Delta w$ represents the over-etch or under-etch perturbations in waveguide widths, and $\Omega$ indicates the input port selection, which now dictates the type of optical operation applied on the input signal. Consequently, the target transfer function $T_{target}(\Omega,\lambda)$ is now also a function of $\Omega$. For this more general figure of merit, $Q$ is the updated total number of combinations of all wavelengths, etch-offsets, and input port specifications.

This formulation allows us to design networks with more complex relationships between input-output pairs while simultaneously achieving tolerance against fabrication variations. We showcase this capability by designing a fabrication-tolerant photonic network with two inputs and three outputs, with a combined power splitter functionality, as illustrated by the device schematic in Fig. 6a. The target functionality for

this device is configured such that light entering the center input is separated equally between the three outputs (1/3, 1/3, 1/3), whereas the light entering the top input is separated equally between only the top and bottom output ports (1/2, 0, 1/2) throughout the entire C-band. This network is constructed from four consecutive layers of interferometers as shown, resulting in a total footprint of $8 \times 320\ \mu m^2$.

For analysis of fabrication-tolerant design capability, we demonstrate the performance of networks designed both without and with tolerance to fabrication errors. The evolution of figures of merit throughout the optimization processes are plotted in Fig. 6b, c. In Fig. 6c, five different $\Delta w$ offsets (-20 nm, -10 nm, 0 nm, 10 nm, 20 nm) were considered. In this fabrication-tolerant design, as the optimizer takes into account not a single network but five different networks simultaneously, the resulting figure of merit effectively includes optimizing the transfer function of a total of $5 \times 4 = 20$ MZIs. From this perspective, device optimization under fabrication errors inherently involves scaling to a larger number of interferometers, simply by the nature of this target functionality. While scaling in such artificial dimensions has obvious practical differences from spatial scaling in network depth or width, the resulting fabrication tolerance capability can be considerably more important for usability in application settings. For this specific example, the final figures of merit for the ideal and fabrication-tolerant networks were $1 \times 10^{-5}$ and $5 \times 10^{-5}$, respectively. As anticipated, the fabrication-tolerant device yields slightly

a)

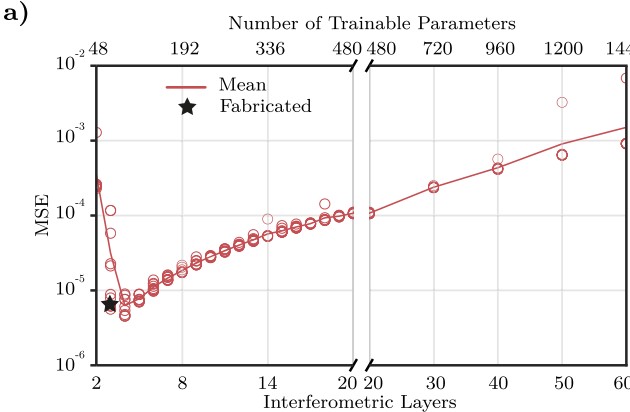

b)

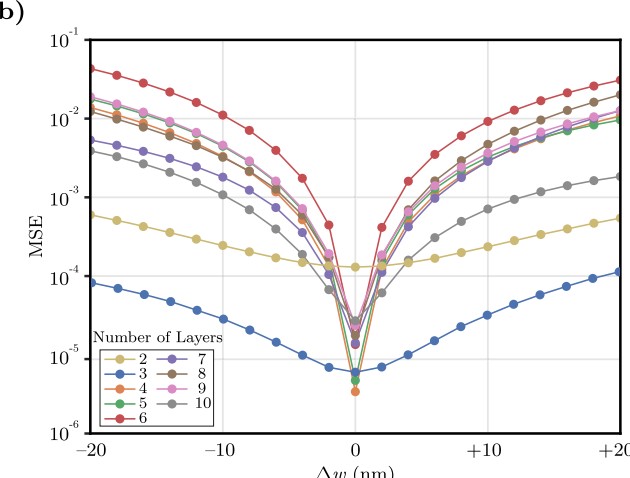

**Fig. 5 | Influence of network size on final device performance. a** Performance of the 50/50 power splitter deep photonic network as a function of the number of interferometric layers, which directly controls the number of trainable parameters. The plotted mean squared error includes propagation loss in the directional couplers and the S-bends extracted from their 3D-FDTD simulations. For each network size, ten different randomly-initialized devices are optimized and depicted with red circles. **b** Robustness of device performance against fabrication-induced variations with number of layers from $M = 2$ through $M = 10$. While increasing the number of interferometric layers initially provides better-performing devices under ideal fabrication conditions ($\Delta w = 0$), longer devices perform worse under significant fabrication variations due to accumulating phase errors.

-20 nm to 20 nm. While the ideal device clearly achieves a better absolute error under no fabrication errors ($\Delta w = 0$), the fabrication-tolerant network demonstrates larger tolerances by maintaining a significantly lower figure of merit in case of non-zero $\Delta w$. These results also demonstrate the practicality of our design framework for integration in a wide variety of applications and fabrication platforms, by giving system designers a choice in the final selection between different designs, which can be influenced by the specific fabrication procedures used.

## Discussion

Our design framework provides a computationally efficient, physically accurate, and systematic methodology for creating deep photonic network architectures for on-chip arbitrary optical systems. The design framework is also capable of extended functionality for specific output configurations enabling band-pass filters with different bandwidths (Supplementary Section 6) as well as devices with constant dispersion profiles (Supplementary Section 7). For all of our demonstrations, while we only focused on silicon-based devices, the presented methodology is applicable in a wide variety of material platforms and spectral applications. Currently, each MZI in our deep photonic networks is 80 µm long and 4 µm wide, due to size of the directional couplers (Supplementary Section 1) and 10 µm-long custom tapers. Depending on the network width and depth, these dimensions result in footprints from 960 µm² to 1920 µm² for our experimentally demonstrated devices, which are either consistent with or smaller than those of integrated interferometer meshes in literature. These include programmable[3,7,35,36] photonic information processors whose responses also require additional electrical system stability, as well as meshes specifically targeting compact network structures with typical reported optical subsystem footprints ranging from 0.025 mm² to multiple mm² (not including electrical interfacing, metal routing, or contacts)[6,37–39]. Moreover, our design framework also uniquely benefits from its ability to effectively combine multiple functional devices into a single photonic network, as demonstrated by the results in Fig. 6. Even though more complex optimization objectives may require longer devices with inherent size limitations (Supplementary Section 8), such multi-functional integration presents an additional and unique avenue towards achieving much higher on-chip integration density, while still maintaining broad optical operation bandwidths.

Our 50/50 and 75/25 power splitters demonstrate simulated 1dB bandwidths of over 200 nm, and experimentally measured 1dB bandwidths as wide as the entire measured spectrum of 120 nm. Both devices operate with insertion losses below 0.61 dB. In comparison to previous experimental demonstrations[40–50], these metrics represent the state-of-the-art performance in bandwidth, and illustrate comparable performance in insertion loss. Likewise, our duplexer demonstrates better experimental performance than devices with similar functionality[9,14,51,52], with less than 0.66 dB insertion loss, flat-top transmissions at both outputs, and a cutoff wavelength shift of only 5 nm. Despite the operation bandwidth reaching over 120 nm, this achieved spectral shift is also similar to reported metrics from literature where specific cutoff wavelengths for resonators, filters, or duplexers typically deviate from their targets by several nm[9,14,51–54]. Depending on specific application requirements, this shift can be compensated through standard thermal tuning mechanisms[55,56]. Similarly, based on application needs, the roll-off between the two bands may also be improved by optimizing with a tighter spectral placement of transmission targets shown in Fig. 3(i). These previously reported splitters and duplexers range approximately between 5 µm² and 6 mm² in footprint, depending on their operation principles and constituent waveguide structures. While some of these previous demonstrations using basic ring resonators[53,54,57–59], Y-junctions[44,50,60,61], and subwavelength grating waveguides[43,45,49,62] can achieve functionally similar operation within smaller footprints than our deep photonic

worse performance as evidenced by the larger figure of merit. Moreover, increased complexity due to the consideration of multiple objectives for this device results in a greater number of iterations needed for convergence. However, despite doubling the number of iterations, we note that total optimization time recorded only increases by less than 5 seconds, underscoring the computational efficiency of the design framework (more details can be found in Supplementary Section 4). A comparison of performance for the two devices is shown in Fig. 6d, e. Despite the slightly larger figure of merit for the fabrication-tolerant network, both devices demonstrate near-perfect transmissions under ideal fabrication conditions. However, under non-zero $\Delta w$ offsets, the fabrication-tolerant device maintains much flatter transmission spectra on all of its output ports throughout the entire C-band. This result demonstrates the ability of deep photonic networks to achieve more complex and multi-functional capabilities while simultaneously enabling much better robustness against fabrication errors across all output ports, for all objectives, through the entire design spectrum. We quantify these built-in fabrication tolerance capabilities further in Fig. 6f by plotting the mean squared error as a function of $\Delta w$ for over-etch and under-etch scenarios ranging from

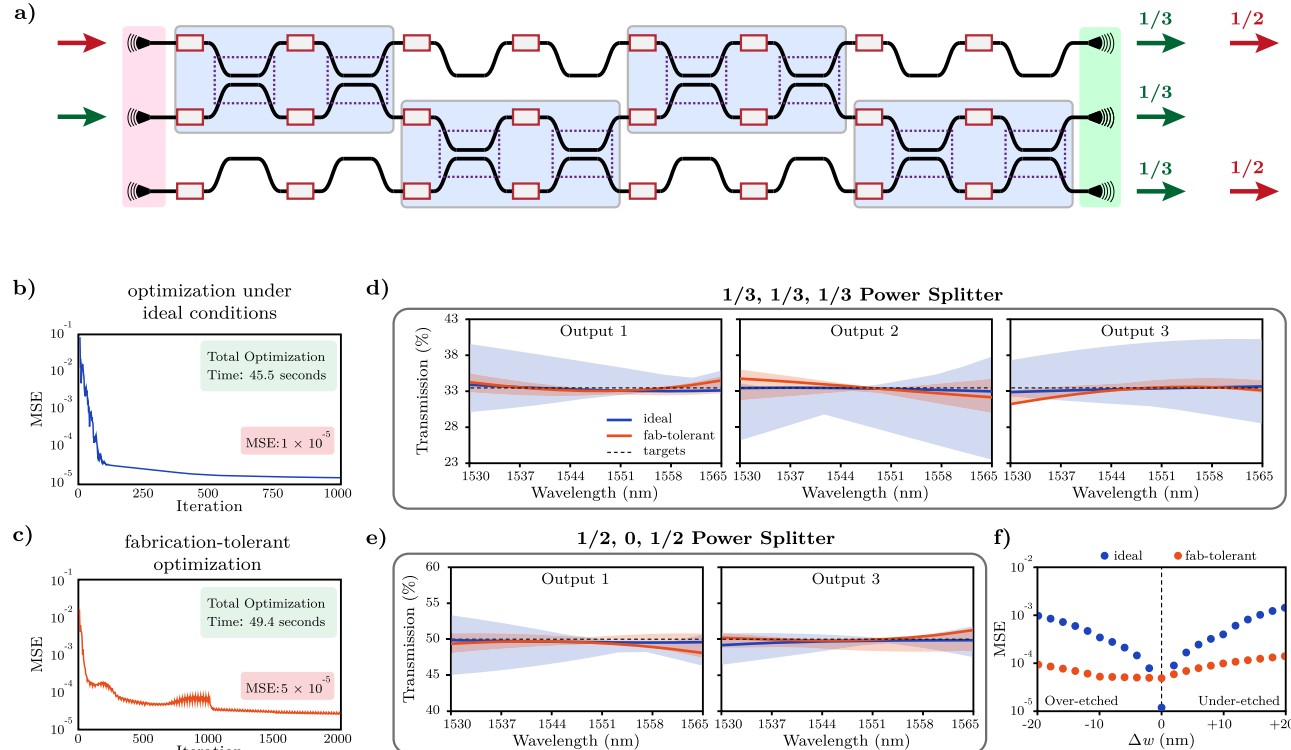

**Fig. 6 | Multi-objective optimization of a deep photonic network with multiple different power splitter capabilities and tolerance against fabrication variations. a** Schematic drawing of the 2-input, 3-output deep photonic network with 4 layers of interferometers. The network acts as a 1 × 3 splitter for the center input (green arrows), and as a 1 × 2 splitter for the top input (red arrows). The mean squared error throughout optimization of the network with 144 trainable parameters for **b** the ideal device and **c** the fabrication-tolerant device. Convergence is achieved in less than one minute for both devices. **d, e** Transmission at the designated output ports of ideal (blue) and fabrication-tolerant (orange) devices as a function of wavelength. Solid lines indicate performance under no fabrication variations, and shaded areas indicate deviation from this performance in case of over-etch and under-etch variations of up to 20 nm. **f** Mean squared error subject to over-etch and under-etch variations for both devices. With ±20 nm modification of waveguide widths, the resulting error is more than 10x better for the fabrication-tolerant device than the ideal device.

networks, their capabilities remain limited to well-defined and fundamental operations with potentially narrower operation bandwidths. Even for devices obtained with free-form inverse-design techniques[9,13,14,40,45,63], the types, complexity, and bandwidth of possible optical operations are practically restricted by the inherent computational difficulty of addressing complicated objectives that require greater degrees of freedom and larger device sizes. In contrast, our networks naturally scale to a greater number of input-output pairs, with little change in their computational optimization performance (Supplementary Section 4). As a result, deep photonic networks allow for a wide and diverse array of demonstrated functional capabilities as complex as arbitrary, multi-functional, and inherently fabrication-tolerant power splitters, duplexers, band-pass filters, and dispersion compensators. As such, these networks not only advance the state-of-the-art in device performance, but also create new pathways for custom photonic system solutions.

In summary, our design framework enables highly scalable implementations of arbitrary transfer functions on-chip, by casting the problem of photonic design as a constrained optimization problem with inherent fabrication compatibility. By integrating accurate waveguide parameters and 3D-FDTD simulations into a physics-informed machine learning architecture, this methodology enables rapid yet accurate simulations of photonic devices and their scalable optimization. Our modular network design allows for a large number of degrees of freedom through custom layers of MZIs, allowing for complex photonic functionality, and therefore presents a tractable path forward for the design of large-scale integrated photonic systems. Moreover, as our computational design framework keeps track of complete phase

information through the individual network components, it allows for the design of photonic networks with specific phase and dispersion profiles as a part of their target functionality. Due to the availability of rapid individual device simulations, our framework can also be configured to enable future designs with on-chip amplifiers and lasers[64,65], electrically-interfaced modulators and detectors[66], as well as structures with robustness against fabrication-induced variations[67]. These capabilities present exciting novel directions in the design of photonic components with arbitrary transfer functions for use in next generation optical communication applications, neuromorphic photonic information processors, and medical/biological sensing.

## Methods
### Numerical simulations
The effective indices of silicon strip waveguides were extracted using Silicon Photonics Toolkit[33], an automatic differentiation-compatible open-source software package for the design of integrated photonic structures. This package enables fast lookup and evaluation of waveguide parameters on the 220 nm SOI platform, which is critically important for the rapid and scalable evaluation of our optical transfer functions. In our deep photonic networks, optical responses of the other components including directional couplers and waveguide bends were extracted from 3D-FDTD simulations performed with a maximum spatial discretization of 17 nm in all three dimensions. These responses including both amplitude and phase information were then linearly interpolated at 1000 wavelengths between 1.2 μm and 1.7 μm. The resulting interpolations were implemented as automatic differentiation-compatible lookup functions, and used

during the performance evaluation of the constructed photonic networks.

## Numerical optimization framework

Our deep photonic network optimization framework was built on an open-source, end-to-end deep learning library[68], enabling the use of state-of-the-art machine learning software constructs as well as access to modern hardware accelerators including GPUs and TPUs. In this framework, we model each interferometric structure as part of a physics-informed artificial neural network, and evaluate the amplitude and phase profiles of the transfer functions between each input/output pair using the automatic differentiation-compatible functions described above. This modular and highly parallelizable architecture allows for serial, parallel, or even residual types of connections between interferometric layers, which can also be used for constructing more complicated network topologies. The trainable parameters of our networks are iteratively optimized using adaptive moment estimation[34]. During the optimizations, the learning rate was progressively reduced from $3 \times 10^{-3}$ to $10^{-4}$ for ease and speed of convergence. For the design of the power splitters and the spectral duplexer, we used batch sizes of 32 and 21, respectively. A relative convergence was used for the stop condition of optimizations (see Supplementary Section 4 for details). All optimizations were performed using a single Tesla V100 GPU.

## Device fabrication

After optimization, the final designed devices were converted to mask layouts using capabilities implemented in our design framework, through an open-source layout construction software library[69]. Grating couplers were added at the inputs and outputs of the network in order for on- and off-chip light coupling. The devices were fabricated using standard 193 nm CMOS photolithography techniques on the SOI platform with a 220-nm-thick silicon device layer through IMEC's multi-project-wafer foundry service.

## Experimental measurements

For experimental characterization of deep photonic networks, our measurement procedures include standard steps to remove any losses due to on- and off-chip coupling of optical signals through grating couplers such as reflections[70] or potential mismatches between fiber, grating, or waveguide modes[71]. Our reported insertion losses refer to only the additional losses through the photonic networks themselves, after these grating coupler losses have been removed. The coupling losses have been measured at four separate fiber zenith angles between 8° and 14°, using grating coupler test structures on the same chip as the measured deep photonic networks, and then combined together in order to accurately characterize as wide a measurement bandwidth as possible. All measurements have been performed using a continuous-wave tunable laser source (Santec TSL-710), an optical power meter (Santec MPM-210), and a polarization controller. The tunable source was operated using a wavelength sweep from 1480 nm to 1600 nm with a sampling rate of 40 ps to obtain the transmission characteristics of the measured structures. The spectral oscillations in our experimental measurements indicate presence of well-known Fabry-Perot interference due to reflections at fiber-to-chip interfaces[72]. These reflections are an inherent result of characterizing the devices on their own, with grating couplers directly connected to the inputs and outputs of our deep photonic networks. As parts of a larger photonic system, the networks can be directly connected by waveguides to other upstream and downstream on-chip devices, eliminating potential reflections at the grating interfaces and any associated spectral oscillations.

## Data availability

The data that support the findings within this manuscript are available from the corresponding author upon request.

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

## Acknowledgements

This work was supported by the Marie Sklodowska Curie Fellowship (number 101032147) through the Horizon 2020 program of the European Commission, and by The Scientific and Technological Research Council of Turkey (grant number 119E195), both awarded to E.S.M.

## Author contributions

E.S.M. conceived the idea of deep photonic networks. A.D.V. created the design framework with simulation, optimization, and layout capabilities. A.N.A. and K.G. developed and revised separate modules of the design framework. A.N.A. designed and simulated the individual devices. K.G.

finalized the mask layout for fabrication. A.N.A. performed the experimental characterization of the devices; and K.G. assisted the setup and experiments. E.S.M. supervised and coordinated the research. A.N.A. and E.S.M. wrote the manuscript with contributions from all co-authors.

## Competing interests

A.N.A., A.D.V., K.G., and E.S.M. have filed a patent application in Turkey (2023/012306), and are in the process of filing a worldwide patent application for the photonic design framework as described in this work.
