## [Peer Review File · Nature Communications]

Deep Photonic Network Platform Enabling Arbitrary and Broadband Optical FunctionalityREVIEWER COMMENTS

Reviewer #1 (Remarks to the Author):

The authors propose a new approach to the creation of integrated optical circuits able to reproduce arbitrary transfer functions. I really appreciate the manuscript, and the proposed approach is, to the best of my knowledge, novel and promising.

The text is easy to read, and the supplementary material contains several indications about how to reproduce the results.

Nevertheless, I think that three aspects should be better investigated/discussed in the text in order to give a clearer understanding of the possibilities and limits of this approach:

1) Fabrication Tolerances:

I really appreciate the fact that the authors discuss the impact of fabrication tolerances on the system performance, but then the natural question is whether the system can be used to design the circuit not only considering the ideal transfer function, but also considering the possible fabrication errors. In certain cases (depending on fabrication facilities and optical structure) when considering two designs, one yielding the highest match with the desired transfer function, while the other giving a slightly worse performance but offering larger tolerances, the final selection can be largely influenced by the fabrication procedure.

2) More complex functions:

While the reported examples are very clear, I think that it would be important to add (in the text or at least in the supplementary material) a few cases, like:

- Creation of an optical bandpass filter with different bandwidth (e.g. 50 GHz and 200 GHz) and high extinction ratio, for optical communication systems.
- Creation of an all-pass filter introducing a constant dispersion, i.e. a group delay growing linearly with the wavelength, so as to show the ability of the system to control the phase of the signal (as needed for dispersion compensators)

3) Impact of large building blocks:

While the approach is extremely promising, one evident drawback with respect to other approaches (more time-consuming and computation intensive) is that the fundamental building blocks in this case are quite large. It would be important to compare the required chip area with respect to other components reported in the literature, and to try to give an idea of what is the maximum complexity of the achievable functions, which could be limited by chip-area occupation, losses, or errors accumulation between layers.

Reviewer #2 (Remarks to the Author):

Authors introduce a design methodology to achieve an arbitrary transfer function from a network of connected static Mach-Zehnder interferometers in integrated photonics. The MZI are composed of 3dB splitters and phase sections which are optimized individually. The phase sections are waveguides with arbitrary spatial modulation profile of their width. The transfer function is computed using the transfer matrix formalism (only for forward waves) and the phase is modelled as a differentiable function of the parameters. There comes the main idea, which is to use backpropagation in order to compute the gradient of the error with respect to the parameters. The optimisation procedure is therefore numerically efficient, which satisfies a requirement for the realisation of scalable photonic circuits.

Three functions have been implemented for an experimental demonstration: 50:50, 25:75 power splitters and spectral duplexer using a chain of cascaded MZI. The experimental curves are close enough to the target and observed deviations consistent with fabrication tolerances, as shown by the analysis.

Here are my comments

1) the main claim is "scalable implementation of arbitrary functions" and "complex photonic functionality". While I agree that the approach is numerically efficient and, to be clear, I appreciate it, I do not see scalability nor complexity in the examples shown here: the number of MZI is limited and, besides, the cascaded geometry considered is much simpler than the mesh shown in fig 1. Perhaps it is just a matter of choosing the right words. Or demonstrating a specific phase and dispersion profile as authors suggest in the conclusions.

2) Comparison with the literature must be made in order to claim that experimental results are state of the art. This should also include footprint as a criterion. Also, to show that deviations are "minimal" some references should be given. For instance, the transition in the duplexer spectral function is not very sharp, therefore spectral deviation may appear minimal.

When quoting "excess losses", do authors refer to on-chip losses, thereby removing coupling losses? How do they calibrate the measurement? In fig. 4a,b, there is a structure of oscillations which might be due to Fabry Perot interference due to residual reflection (at the fiber couplers). I would expect some comments here.

3) I think that the description of the mathematical model should be improved. For instance eq. 2 describes a chain of MZI (as in fig 3) but not the more general mesh described in fig 1. As a minor point, are authors sure that "scattering matrix" is appropriate here? Also, neglecting reflection, although plausible and desired, needs some justification. The important assumption is that the arbitrary profiles do not induce any reflection, which might deserve some comment. Besides, did authors measure the reflected light in their circuits?

Overall, the manuscript is well prepared, the quality of the figures is high and the methodology is sound. I suggest authors to improve the introduction which I found not very focused and not particularly useful to describe the scope and the purpose of this work. It could be more succinct in my perspective.

Were authors to demonstrate an example where complexity and scalability are apparent, with respect for instance to what is already available (for instance the reprogrammable photonic circuits implementing an unitary matrix transform with a fairly large number of channels), then they would make a strong case for publication. At this stage I am not yet convinced the results are up to the level of Nature Communication

Reviewer #3 (Remarks to the Author):

Optimization techniques are now the standard in photonic component design. Previous work in this area includes [1-3]. There are multiple works that specifically use backpropagation for photonic device design.

The authors used optimization techniques to aid in the design of three integrated photonic components (with specific spectral responses) that were fabricated. In addition, they developed a simulation framework for chaining together these components and creating "meta" devices. The optimizer, which leverages backpropagation, is controlling "trainable widths" that correspond to the physical geometry of the device. Overall, the authors have produced a design/simulation framework

for multi-port linear optical photonic systems that optimizes over device parameters.

The paper is well written and understandable due to its clear organization. However, the amount of introductory material is unnecessary. The components produced by this method of chaining devices together into "deep photonic networks" did not produce record-breaking results. This is in contrast to the work shown in [2] and I believe publication of this manuscript in Nature Communications would require not only a new method, but a record breaking device/subsystem that resulted from its use. The components demonstrated here are impractically large for industrial applications and would suffer significantly from phase disorder introduced by side wall roughness (requiring phase tuning post fabrication to yield the correct transfer matrix).

Overall, I believe this is solid, scientific work but it does not represent a breakthrough that would justify publication in this journal.

[1] Liu, Zhaocheng, et al. "Tackling photonic inverse design with machine learning." *Advanced Science* 8.5 (2021): 2002923.

[2] Molesky, Sean, et al. "Inverse design in nanophotonics." *Nature Photonics* 12.11 (2018): 659-670.

[3] Zhang, Yi, et al. "A compact and low loss Y-junction for submicron silicon waveguide." *Optics express* 21.1 (2013): 1310-1316.

Responses to Reviewer Comments

We thank the reviewers and the editor for their constructive comments and suggestions. We have revised the manuscript accordingly and have provided detailed responses to each comment below. In the following, we present our point-by-point replies (in blue) to the reviewers' comments (in black), as well as the specific actions taken in the revision of our manuscript (in red).

————Reviewer #1————

Reviewer #1:

“The authors propose a new approach to the creation of integrated optical circuits able to reproduce arbitrary transfer functions. I really appreciate the manuscript, and the proposed approach is, to the best of my knowledge, novel and promising. The text is easy to read, and the supplementary material contains several indications about how to reproduce the results. Nevertheless, I think that three aspects should be better investigated/discussed in the text in order to give a clearer understanding of the possibilities and limits of this approach.”

Our Response:

We appreciate Reviewer #1's comments about the novelty and potential for our work, and for acknowledging our efforts in the organization/presentation of the manuscript. We also believe that the design platform can perform a critical role for the PIC design industry through the design of different modules and systems. We have addressed the three aspects they mentioned in the following sections below.

Reviewer #1:

“1) Fabrication Tolerances:

I really appreciate the fact that the authors discuss the impact of fabrication tolerances on the system performance, but then the natural question is whether the system can be used to design the circuit not only considering the ideal transfer function, but also considering the possible fabrication errors.

In certain cases (depending on fabrication facilities and optical structure) when considering two designs, one yielding the highest match with the desired transfer function, while the other giving a slightly worse performance but offering larger tolerances, the final selection can be largely influenced by the fabrication procedure.”

Our Response:

It is well known that despite the demonstrated performance of any integrated photonic device, the tolerance to fabrication variations remains a critical concern for all scalable applications. Especially for practical use cases in industrial systems (as also mentioned by Reviewer #3), designers face challenges in making optimal decisions in the presence of potentially conflicting trade-offs between achieving ideal functionality versus accommodating fabrication errors. In this regard, we appreciate Reviewer #1's and also Reviewer #3's insightful comments, which have prompted us to demonstrate more advanced capabilities of our design framework, as we detail in the following paragraphs.

In addition to creating a single device with a single optical functionality, our design framework is also capable of optimizing multiple different objectives simultaneously. We use this particular capability to design photonic devices with built-in fabrication tolerance. Specifically, we simultaneously optimize the optical response of multiple different versions of a deep photonic network, each version being the result of a different over-etch or under-etch scenario, thus considering multiple different variations in waveguide widths that may arise from fabrication errors. By integrating this multi-objective approach, our framework can evaluate a combined figure of merit including the device performance not only under ideal conditions, but also with possible fabrication errors.

While this approach outlines how our networks can be designed by taking fabrication errors into account, the use of multi-objective optimization has broader implications for the utility and functionality of deep photonic networks. This fact also relates to a relevant comment by Reviewer #2, who emphasized scalability and complexity of the demonstrated optical capabilities. In this regard, we note that the multi-objective approach can be generalized as a means of designing networks with multiple, physically different functions with more complex functionality as well as scalability across more input-output pairs. For instance, a single photonic network can be configured as a combination of multiple different power splitters, in which the optical response depends on which one of the input ports receives the optical signal. In this case, we define a more general figure of merit as a mean squared error including all possible combinations of fabrication variations and input ports as

$$J(x) = \frac{1}{Q} \sum_{\Omega} \sum_{\Delta w} \sum_{\lambda} |T_{\text{calculated}}(\Omega, \Delta w, \lambda, x) - T_{\text{target}}(\Omega, \lambda)|^2$$

where the width offset parameter Δw represents the over-etch or under-etch perturbations in waveguide widths, and Ω indicates the input port selection, which now dictates the type of optical operation applied on the input signal. Consequently, the target transfer function $T_{\text{target}}(\Omega, \lambda)$ is now also a function of Ω . For this more general figure of merit, Q is the updated total number of combinations of all wavelengths, etch-offsets, and input port specifications.

This formulation allows us to design networks with more complex relationships between input-output pairs while simultaneously achieving tolerance against fabrication

variations. We showcase this capability by designing a fabrication-tolerant photonic network with two inputs and three outputs, with a combined power splitter functionality, as illustrated by the device schematic in Fig. 6(a). The target functionality for this device is configured such that light entering the center input is separated equally between the three outputs ($1/3, 1/3, 1/3$), whereas the light entering the top input is separated equally between only the top and bottom output ports ($1/2, 0, 1/2$) throughout the entire C-band. This network is constructed from four consecutive layers of interferometers as shown, resulting in a total footprint of $8 \times 320 \mu\text{m}^2$.

FIG. 6. Multi-objective optimization of a deep photonic network with multiple different power splitter capabilities and tolerance against fabrication variations. (a) Schematic drawing of the 2-input, 3-output deep photonic network with 4 layers of interferometers. The network acts as a 1×3 splitter for the center input (green arrows), and as a 1×2 splitter for the top input (red arrows). The mean squared error throughout optimization of the network with 144 trainable parameters for (b) the ideal device and (c) the fabrication-tolerant optimization. Convergence is achieved in less than one minute for both devices. (d), (e) Transmission at the designated output ports of ideal (blue) and fabrication-tolerant (orange) devices as a function of wavelength. Solid lines indicate performance under no fabrication variations, and shaded areas indicate deviation from this performance in case of over-etch and under-etch variations of up to 20 nm. (f) Mean squared error subject to over-etch and under-etch variations for both devices. With ± 20 nm modification of waveguide widths, the resulting error is more than 10x better for the fabrication-tolerant device than the ideal device.

For analysis of fabrication-tolerant design capability, we demonstrate the performance of networks designed both without and with tolerance to fabrication errors. The evolution of figures of merit throughout the optimization processes are plotted in Fig. 6(b) and 6(c). In Fig. 6(c), five different Δw offsets (-20 nm, -10 nm, 0 nm, 10 nm, 20 nm) were considered. In this fabrication-tolerant design, as the optimizer takes into account not a single network but five different networks simultaneously, the resulting figure of merit effectively includes optimizing the transfer function of a total of $5 \times 4 = 20$ MZIs. From this perspective, device optimization under fabrication errors inherently involves scaling to a larger number of interferometers, simply by the nature of this target functionality. While scaling in such artificial dimensions has obvious practical differences from spatial

scaling in network depth or width, the resulting fabrication tolerance capability can be considerably more important for usability in application settings.

For this specific example, the final figures of merit for the ideal and fabrication tolerant networks were 1×10^{-5} and 5×10^{-5} , respectively. As anticipated, the fabrication tolerant device yields slightly worse performance as evidenced by the larger figure of merit. Moreover, increased complexity due to the consideration of multiple objectives for this device results in a greater number of iterations needed for convergence. However, despite doubling the number of iterations, we note that the total optimization time recorded only increases by less than 5 seconds, underscoring the computational efficiency of the design framework (more details can be found in Supplementary Section 4).

A comparison of performance for the two devices is shown in Fig. 6(d) and 6(e). Despite the slightly larger figure of merit for the fabrication-tolerant network, both devices demonstrate near-perfect transmissions under ideal fabrication conditions. However, under nonzero Δw offsets, the fabrication-tolerant device maintains much flatter transmission spectra on all of its output ports throughout the entire C-band. This result demonstrates the ability of deep photonic networks to achieve more complex and multi-functional capabilities while simultaneously enabling much better robustness against fabrication errors across all output ports, for all objectives, through the entire design spectrum. We quantify these built-in fabrication tolerance capabilities further in Fig. 6(f) by plotting the mean squared error as a function of Δw for over-etch and under-etch scenarios ranging from -20 nm to 20 nm. While the ideal device clearly achieves a better absolute error under no fabrication errors ($\Delta w = 0\text{nm}$), the fabrication-tolerant network demonstrates larger tolerances by maintaining a significantly lower figure of merit in the case of nonzero Δw . These results also demonstrate the practicality of our design framework for integration in a wide variety of applications and fabrication platforms, by giving system designers a choice in the final selection between different designs, which can be influenced by the specific fabrication procedures used.

For multi-objective optimizations, such as the example shown in Fig. 6, the specified objectives are all simulated simultaneously, where all physical simulations at different wavelengths, using different inputs, and for different etch-offsets are processed in parallel. This computational efficiency allows the optimization framework to scale well for the design of fabrication tolerant devices, and devices with target transfer functions across a large number of input-output pairs. As expected, the fabrication-tolerant optimization in Fig. 6(c) exhibits increased complexity due to the consideration of multiple objectives, resulting in a higher number of iterations needed for convergence in comparison to optimization under ideal conditions shown in Fig. 6(b). However, despite the doubling of the number of iterations from 1000 to 2000, we note that the total optimization time increased only by less than 5 seconds (from 45.5 s to 49.4 s), underscoring the efficiency of our framework. This can also be attributed to the initial

part of the optimization workflow used for JIT compilation of required mathematical operations, which remains independent of the number of iterations. As such, each additional iteration taking a few milliseconds adds minimal computational time to the overall optimization process.

These findings demonstrate more advanced capabilities of our framework and its ability to address the critical aspect of fabrication tolerances in the design of photonic circuits without sacrificing computational efficiency.

Action Taken:

We have included these new results in multiple different sections throughout the main text and our supplementary materials:

- A new section titled “Multi-Objective Design Capabilities” and the new Fig. 6 were added with the details of our combined power splitter device explained above.
- We also added the last paragraph of our explanation regarding the computational efficiency in case of multi-objective optimization in Supplementary Section IV.

Reviewer #1:

“2) More complex functions:

While the reported examples are very clear, I think that it would be important to add (in the text or at least in the supplementary material) a few cases, like:

- Creation of an optical bandpass filter with different bandwidth (e.g. 50 GHz and 200 GHz) and high extinction ratio, for optical communication systems.”

Our Response:

We greatly appreciate Reviewer #1's suggestion to extend our designs to illustrate more complex functions. Our device above, with the combined power splitter functionality and simultaneous fabrication-tolerance, is one possible example demonstration with significantly more advanced functionality. In addition to this example, we can of course demonstrate other devices including band-pass filters for particularly important applications like optical communications.

Our deep photonic network design infrastructure is also capable of arranging transmission objectives in order to create bandpass filters with high extinction-ratios and sharp spectral features. In order to achieve these metrics that are typically required in communications applications, we expand the deep photonic network structure with longer delays in the form of spirals as shown in Fig. S7(a). This structure enables

band-pass optical transfer functions that are otherwise typically achieved with combinations of ring/disk resonators and/or MZIs [21-26]. The updated simulation of optical response for such a network also includes modular integration of lengths of these spiral waveguides as trainable parameters alongside the existing widths and lengths for the custom tapers. The phase acquired through this waveguide is a function of its trainable length, as illustrated in Fig. S7(b). For this specific application, each network consists of a 1-input 2-output configuration with the labeled through and drop port outputs.

FIG. S7. Design of optical band-pass filters with the deep photonic network architecture. (a) Schematic drawing of the network structure where longer spiral waveguides have been added in each Mach Zehnder interferometer for achieving band-pass functionality. (b) The phase accumulated in the spiral waveguide as a function of wavelength is calculated using its trainable length L_s and the default width of $w_{\text{default}} = 450$ nm. The transmission response at through and drop ports of (c) a 200 GHz band-pass filter with 6 layers of MZIs (150 trainable parameters), (d) a 100 GHz band-pass filter with 9 layers of MZIs (225 trainable parameters), and (e) a 50 GHz band-pass filter with 10 layers of MZIs (250 trainable parameters including trainable spiral waveguide lengths). The simulated final 3dB bandwidths are 201.90 GHz, 96.09 GHz, and 50.51 GHz for the three devices, respectively. Optimization for each one of the filters converges in several hundred iterations, in under three minutes of total computation time. All devices are optimized between 1547 nm and 1553 nm.

Using this structure, we have designed three separate optical band-pass filters at a center wavelength of 1550 nm, with target bandwidths of 200 GHz, 100 GHz, and 50 GHz, whose simulated responses are shown in Fig. S7(c)-(e), respectively. The target bandwidth for each filter was controlled by specifying a range of wavelengths at which the entire optical input was transmitted to the drop port around the center wavelength. Inherently, achieving narrower filter bandwidths while maintaining a good extinction ratio represents a more difficult problem, and therefore, requires greater degrees of optical design freedom. Consequently, 6-layer, 9-layer, and 10-layer networks were used for designing the three filters, in decreasing order of optical bandwidth. The demonstrated bandwidths for the optimized devices were 201.90 GHz, 96.09 GHz, and 50.51 GHz for the three filters shown. At the center wavelength of filter transmission, extinction ratios of -48 dB, -25 dB, and -35 dB were obtained. As indicated by these transmission results, all three filters demonstrate agreement with their target specifications and achieve sufficiently high extinction ratios for communication applications. In contrast to

our power splitter examples, we have included a dB-based specification of optical transmission between the through and drop ports in these band-pass filter demonstrations. From an implementation perspective, this allows the optimizer to target a given maximum cross-talk between the outputs, and modify trainable width and length parameters towards achieving this goal. With this specification, our filters have achieved better than -20 dB cross-talk between the two outputs for the majority of design spectrum.

Before optimization, spiral waveguide lengths were randomly initialized using $L_s = L_{\text{default}} - p_L \cdot \Delta L$, where p_L is a random variable between 0 and 1, and ΔL is a user chosen maximum deviation amplitude from the default, similar to the initialization of the lengths and widths of the custom tapers. After a set of optimization trials with different initializations, as is commonplace in machine learning model training [5-7], in order to achieve gradually narrower filter bandwidths, L_{default} of 50 μm , 80 μm , and 155 μm were used, respectively for the three devices. As anticipated, longer spiral lengths yielded photonic networks that are more suitable as narrow-bandwidth filters in our optimizations. However, the exact dependence of filter bandwidth on waveguide lengths remains a more complex function of optical power transmitted through each arm, which is influenced by the phase relationships between all custom tapers and spirals. The maximum final optimized spiral waveguide lengths remained below 55 μm for the 200 GHz filter, and 160 μm for all other filters designed. Due to the compact structure of the spiral geometry, their placement can be planned in rectangular blocks adjacent to one of the interferometer arms as shown, with only a minor impact on the overall device footprint.

These examples demonstrate the capability of our framework to efficiently design optical band-pass filters with varying bandwidths and high extinction ratios, further expanding the practicality and applicability of our approach in the context of optical communication systems.

Action Taken:

We have added the following paragraphs and the new Fig. S7 of the band-pass filter results in the new Supplementary Section VI of our submission.

Reviewer #1:

“Creation of an all-pass filter introducing a constant dispersion, i.e. a group delay growing linearly with the wavelength, so as to show the ability of the system to control the phase of the signal (as needed for dispersion compensators)”

Our Response:

We appreciate Reviewer #1's suggestion to illustrate our framework's capability in designing dispersion compensating devices, which essentially control signal phase to achieve constant dispersion. This is something we had already suggested in our discussion as a potential use case for deep photonic networks. Below, we detail the design of these devices and provide example demonstrations.

Traditionally, on-chip constant dispersion is achieved using components like cascaded ring resonators [27-29] or Bragg gratings [30-32] due to highly linear group delay within their transmission/reflection bandwidths. At the wavelengths within their acceptable dispersion compensating bandwidth, these devices can inherently achieve near unity transmission, by the nature of their operation principles. A similar functionality can be accomplished by deep photonic networks; but this requires formulation of a combined objective with separately specified transmission and dispersion targets, in a fashion similar to the multi-objective designs we demonstrated. To do so, we separate the complex output of a photonic network into its amplitude and phase components, enabling calculation of transmission and group delay (as well as dispersion) results at each output port. For calculation of dispersion, the first and second derivatives of phase recorded and unwrapped at respective output ports are computed, using twice differentiable interpolations of effective index and directional coupler response. Compared to our transmission-only objectives, here we use a narrower wavelength spacing of 0.1 nm in order to avoid potential undersampling issues during unwrapping of this phase. We then define a figure of merit including all of these metrics to quantify the difference between target and calculated transmission and dispersion responses as

$$J(x) = \frac{1}{Q} \sum_{\lambda} \left((1 - \eta) |T_{\text{calculated}}(\lambda, x) - T_{\text{target}}(\lambda)|^2 + \eta |D_{\text{calculated}}(\lambda, x) - D_{\text{target}}(\lambda)|^2 \right)$$

where T and D indicate the transmission and dispersion responses of the network as a function of wavelength λ and trainable parameters x , and we defined an additional scaling parameter η . The control of this parameter η allows the designer to specify a relative weight between the transmission and dispersion objectives. We find experimenting with this parameter to be particularly useful in order to optimize for a certain amount of dispersion within the desired bandwidth, while still minimizing the power lost to other output ports to achieve low insertion loss (near unity transmission). For optimizing devices with this given figure of merit, we initialize and iteratively update trainable lengths and widths of a deep photonic network as before.

As proof of principle, we demonstrate two networks with a target constant dispersion of 0.5 ps/nm, with bandwidths of 3 nm and 6 nm, centered at 1550 nm, using 7 layers and 9 layers of MZIs, respectively. The geometry of these dispersion-compensating networks is structurally identical to the band-pass filters in Fig. S7(a); but the drop port remains unused as the entire output is collected from the through port in these devices. Here, the spiral waveguides were initialized to be around 47 μm in length, and then were iteratively optimized together with the custom tapers in the photonic network. The

resulting group delays are plotted in Fig. S8(a) and S8(d), with both devices illustrating linear delay profiles confirming successful device optimizations. The dispersion is calculated from the derivative of this group delay, and plotted for the two devices in Fig. S8(b) and S8(e). The dispersion profiles deviate from the constant target of 0.5 ps/nm by 35-40 fs/nm towards the end of the specified bandwidth. Overall, the narrower-band device achieves better agreement with the dispersion target specified, due to the relative difficulty of achieving constant dispersion across wider optical bandwidths.

A similar observation can be made regarding the transmission results in Fig. S8(c) and S8(f). The 9-layer device with 6 nm of target constant dispersion spectrum experiences slightly higher insertion loss, especially towards the end of the specified spectral range. As before, this emphasizes the relative difficulty of the target specification as simultaneously satisfying transmission and dispersion objectives across a wider bandwidth represents a more difficult optimization problem. Still, we observe better than -0.4 dB of insertion loss throughout the majority of the operation bandwidth, which is consistent with previous results from the literature [30, 33-35].

For compensating higher amounts of dispersion, multiple copies of the same networks can be cascaded in series, as their insertion loss remains relatively small. In comparison, while the demonstrated amounts of dispersion compensation in literature can exceed 10 ps/nm [30, 34-36], our network structure allows for not only constant anomalous dispersion for dispersion compensating applications, but also specification of arbitrary target dispersion profiles. These capabilities allow for unique design freedom in pulse-shaping [37-39] and quantum information processing applications [40], which typical dispersion compensators cannot provide. Even though cascaded ring resonators technically have the capability to achieve arbitrary dispersion profiles [27], their limitations become apparent when considering that each ring typically offers just two design parameters, thereby severely restricting the degrees of design freedom compared to our architecture.

FIG. S8. Examples of deep photonic networks for dispersion optimization. Simulated group delay as a function of wavelength for (a) a 7-layer deep photonic network with 175 trainable parameters, and (d) a 9-layer deep photonic network with 225 trainable parameters (these parameters include the lengths and widths in custom tapers as well as the lengths of spiral waveguides). The network structure is identical to the one illustrated in Fig. S7(a); but the entire output is collected from the through port. (b), (e) Spectral dispersion profiles of the two optimized devices. Both devices demonstrate near-constant dispersion throughout their optimized bandwidth of 3 nm and 6 nm, respectively. A target constant dispersion of 0.5 ps/nm was used. (c), (f) Transmission through the devices remains better than -0.4 dB for the majority of their optimized spectrum. Some more insertion loss is observed towards the end of the specified spectral range. (GDD: group delay dispersion)

Action Taken:

We have added the above explanations and the new Fig. S8 in the new Supplementary Section VII of our submission.

Reviewer #1:

“3) Impact of large building blocks:

While the approach is extremely promising, one evident drawback with respect to other approaches (more time-consuming and computation intensive) is that the fundamental building blocks in this case are quite large. It would be important to compare the required chip area with respect to other components reported in the literature, and to try to give an idea of what is the maximum complexity of the achievable functions, which

could be limited by chip-area occupation, losses, or errors accumulation between layers.”

We appreciate the reviewer's attention to the resulting footprint of the demonstrated devices and that of the general approach we present. The on-chip footprint is obviously an important factor for designers to keep in mind for planning integrated systems, especially when a large number of components have to be integrated together. Many designers for both photonic and electronic systems think about footprint, particularly in relationship to the achieved capabilities of the resulting devices. Inherently, with advantages and disadvantages of each design methodology, there are tradeoffs between this capability, computational load for device design, and the final system footprints. We discuss these considerations, limitations, and potential implications below, and provide comparative information on the required on-chip area for various optical functionalities for other types of devices and design methodologies.

The physical dimensions of any demonstrated system and its functional capabilities are influenced by fundamental factors inherent in the nature of its constituent components. As such, blocks of cascaded MZIs make up the vast majority of the resulting device footprints for our modular networks. Currently, each MZI in our deep photonic networks is 80 μm long and 4 μm wide, due to size of the directional couplers (Supplementary Section I) and 10 μm -long custom tapers. Depending on the network width and depth, these dimensions result in footprints from 960 μm^2 to 1920 μm^2 for our experimentally demonstrated devices, which are either consistent with or smaller than those of integrated interferometer meshes in literature.

These include programmable [3, 7, 35, 36] photonic information processors whose responses also require additional electrical system stability, as well as meshes specifically targeting compact network structures with typical reported optical subsystem footprints ranging from 0.025 mm^2 to multiple mm^2 (not including electrical interfacing, metal routing, or contacts) [6, 37-39]. Moreover, our design framework also uniquely benefits from its ability to effectively combine multiple functional devices into a single photonic network, as demonstrated by the results in Fig. 6. Such multi-functional integration presents an additional and unique avenue towards achieving much higher on-chip integration density, while still maintaining broad optical operation bandwidths.

Nevertheless, by the general nature of their design, MZI-based photonic networks are larger in size compared to devices obtained with free-form inverse-design techniques [1, 2]. Functionally similar beam splitting or spectral splitting capabilities have also been reported with such inverse design methods, resulting in footprints ranging from approximately 5 μm^2 to 100 μm^2 [11, 15, 41-44]. However, the diversity and bandwidths

of optical capabilities achieved by these methods are limited to more basic operations across narrower bandwidths, due to the computational difficulty in addressing arbitrary design objectives. At the same time, as mentioned in some form by all three reviewers, even though our deep photonic networks demonstrate superior computational efficiency and higher design freedom, there are also practical limitations to their size and functionality, such as those influenced by propagation losses, accumulation of fabrication errors, and phase disorder due to sidewall roughness.

- Of these factors, we expect typical SOI waveguide propagation losses of 2-3dB/cm not to practically introduce any additional design considerations specific to our photonic networks, since any propagation-related losses remain balanced between our symmetrically-placed interferometer arms by design. Using an MZI length of 80 μm , even for the longest, 10-layer device we have shown in Fig. S4, this corresponds to an expected total loss of approximately $3\text{dB/cm} \times 80 \mu\text{m} \times 10 = 0.24 \text{ dB}$, which is acceptable for many on-chip optical applications. It is also important to note that the symmetric design of all interferometer arms in a given layer ensures propagation through an equal number of directional couplers and S-bends for each arm. The expected total loss through these individual components is already extracted from 3D-FDTD simulations, and is included in our analysis shown in Fig. 5 and Fig. S2. Up to a network depth of 10 layers, these plots illustrate that the optimized figures of merit remain below 10^{-4} (in line with the figures of merit obtained for the fabricated devices), indicating no practical influence of the added scattering losses due to the S-bends or directional couplers.
- Accumulation of fabrication errors through cascaded nodes presents an important challenge for any mesh-like structure. For deep photonic networks, since each individual device simulation can be performed on the order of milliseconds, our design framework enables building inherent tolerance to fabrication errors, as already shown in Fig. 6. Through this approach, the custom tapers are automatically optimized by taking into account multiple different under-etched and over-etched versions of the same device. We expect this added capability to provide significant flexibility in selecting the most appropriate designs for a specific task, which may also be influenced by fabrication methods used. However, as we have already discussed, longer devices naturally exhibit worse tolerances to fabrication errors accumulating through multiple interferometric layers (see Fig. 5(b), Fig. S2(b), and Fig. S2(d)). Therefore, for networks deeper than 10 layers, we expect practical limitations towards achieving similar levels of fabrication tolerance, which may require electrical, post-fabrication tuning.

- Longer devices also suffer from potential phase errors due to sidewall roughness in the fabricated waveguides. This is an especially well-known problem in high-index contrast platforms, as the larger index difference results in phase errors greater in magnitude. This is also an important reason behind the preference of certain optical phased array demonstrations toward SiN guiding platforms instead of Si, due to the comparatively lower index contrast and the smaller magnitude of resulting phase errors [45-48]. Using the 220 nm-thick commercially available SOI platform from IMEC [49], we have observed no detrimental effects on device performance due to this phenomenon for our experimentally demonstrated networks in Fig. 4, up to a network depth of 6 layers. In case of phase errors for deeper networks, a similar approach to implementing optical phase arrays in SiN platforms can be undertaken, since our design framework remains generalizable to other waveguide materials or platforms. While the diversity of optical operations and capabilities in design freedom we have shown can provide solutions for many optical applications within less than 10 layers, the presented design framework can be easily generalized to other guiding materials with smaller phase errors. It may also be possible to implement inherent fabrication tolerance to target these phase errors, in a similar fashion to achieving near-optimal optical responses for over-etch/under-etch scenarios discussed above.

In addition to the comparison above, we also note that other structures may be used in place of the directional couplers inside the interferometers to achieve smaller device footprints. This is partly enabled by the design framework's ability to computationally separate the coupler response from that of the custom tapers, to achieve a modular simulation workflow. For instance, these couplers can be replaced with their more compact counterparts like those designed with inverse-design approaches [1, 2]. This can result in reducing the $4 \times 30 \mu\text{m}^2$ directional coupler footprint to typical inverse design device sizes below $4 \times 4 \mu\text{m}^2$ [3, 4]. The resulting network sizes can be improved by a factor of 4-5x, further increasing integration density. Our future research directions regarding these networks already include incorporating such couplers and even leveraging geometrical placement optimization algorithms to further reduce their footprint.

Action Taken:

- We have added the first explanation paragraph in our response above to our discussion.

- In order to compare our achieved functional capability with other design methods, and convey the results better in reference to potential loss or fabrication related limitations, we added the relevant details in a new supplementary section VIII, titled “Comparison and Limitations of Functional Network Capability”.
- Regarding further improvement of network size, we added the last paragraph of our explanation above in Supplementary Section I.

————Reviewer #2————

Reviewer #2:

“Authors introduce a design methodology to achieve an arbitrary transfer function from a network of connected static mach-zehnder interferometers in integrated photonics. The MZI are composed of 3dB splitters and phase sections which are optimized individually. The phase sections are waveguides with arbitrary spatial modulation profile of their width. The transfer function is computed using the transfer matrix formalism (only for forward waves) and the phase is modeled as a differentiable function of the parameters. There comes the main idea, which is to use backpropagation in order to compute the gradient of the error with respect to the parameters. The optimization procedure is therefore numerically efficient, which satisfies a requirement for the realization of scalable photonic circuits.

Three functions have been implemented for an experimental demonstration: 50:50, 25:75 power splitters and spectral duplexer using a chain of cascaded MZI. The experimental curves are close enough to the target and observed deviations consistent with fabrication tolerances, as shown by the analysis.”

We thank Reviewer #2 for carefully reading the details of our work, and emphasizing the computational efficiency of our implementation. Critically, any computer-aided optimization framework has to use an efficient implementation of differentiable physical simulations, in order to be practically useful and generalizable to complex functions and more input-output pairs. As they also noted, this is a key part of our design workflow. Our detailed responses to the specific comments are listed below.

Reviewer #2:

“1) the main claim is "scalable implementation of arbitrary functions" and "complex photonic functionality". While I agree that the approach is numerically efficient and, to be clear, I appreciate it, I do not see scalability nor complexity in the examples shown here: the number of MZI is limited and, besides, the cascaded geometry considered is much simpler than the mesh shown in fig 1. Perhaps it is just a matter of choosing the right words. Or demonstrating a specific phase and dispersion profile as authors suggest in the conclusions.”

Our Response:

We appreciate Reviewer #2 for pointing out their concerns regarding the achievable complexity and scalability of the demonstrated circuits. To specifically address these comments, we have now included demonstrations of multiple added photonic devices including more complex functionality, scalability across multiple dimensions (both with the numbers of input-output pairs, and with an inherent tolerance for fabrication errors), and also devices with expanded functionality for dispersion-based optimization. With these additions, we show cascaded geometries including a larger number of ports similar to the general case illustrated in Fig. 1, and photonic networks reaching a depth of 10 layers. The possibility of some of these capabilities had also been brought up by Reviewer #1 in their comments.

Before explaining these new categories of devices in detail, we would like to point out that much larger networks have already been demonstrated as a part of our results shown in Fig. S3, but have been potentially hidden behind our discussion of computational performance. In Fig. S3, indicated by “8-port randomly-distributed output powers” is a photonic network structurally consisting of 8 inputs and 8 outputs, for which a randomly specified, complex set of output power distributions (for one of the inputs) has been optimized for a total of 32 wavelengths between 1400 nm and 1600 nm. At a depth of 10 layers, this network is constructed from 35 unique MZIs in a cascaded geometry, resulting in a total of 840 trainable widths and lengths, achieving the same convergence criteria as all of our other demonstrations. For completeness of these results, we plot the simulated transmissions at each output port, together with a schematic diagram of this 8-port power splitter in Fig. S4. The target outputs consist of eight different and randomly-selected transmissions indicated by the dashed lines. The simulated final output transmissions as a function of wavelength are plotted with the solid curves for each one of the eight output ports. The results demonstrate good agreement between the target and simulated transmissions, indicating that the design capability of our design framework scales well to both wider and deeper photonic network structures. Moreover, achieving the desired photonic functionality at these scales does not sacrifice computational efficiency, as the entire optimization for this network only takes approximately 12 minutes.

FIG. S4. 8-port deep photonic network with 10 layers, optimized for randomly-selected output powers. (a) Structure of the deep photonic network with a total of 840 trainable width and length parameters. Light injected at input number 4 propagates through 30 of the 35 placed MZIs. (b) Transmission response of eight randomly-selected, broadband transmission objectives specified at 32 evenly spaced wavelengths between 1400 nm and 1600 nm, as plotted by the horizontal dashed lines. The simulated final output transmissions are plotted by the solid curves for each output port.

We also note that while the device in Fig. S4 and all other individual data points plotted in Fig. S3 illustrate networks with larger sets of input-output pairs, realizing practical optical functionalities with the fewest number of interferometric layers remains the

primary goal of our photonic network demonstrations. As we have analyzed in Fig. 5 and Fig. 2S, larger networks suffer more strongly from fabrication-induced changes. Therefore, from a usability perspective, the number of layers is an essential design choice with critical tradeoffs between network capability and fabrication tolerance. Consequently, in our new demonstrations below, while we include examples of devices with a greater number of input-output pairs, we also emphasize scalability in other dimensions including complexity of the optical function specified, and the capability of

operation under potential fabrication errors. For each one of the three specific new categories of devices we included in our manuscript, we detail the contributions below. As our answer includes all three sets of newly demonstrated devices from our response to Reviewer #1's questions, some of the explanations below are inevitably repeated from above.

Device Category 1

To better address scalability across multiple input-output pairs (and also functional complexity), we demonstrate a new device in the newly added section of our manuscript titled "Multi-Objective Deep Photonic Networks", showing a combined power splitter capability using a network with 2 inputs and 3 outputs. This specific demonstration targets the combination of a 1×2 and 1×3 power splitter functionality, in which two input ports of the device perform two different functions, using the same three outputs. This allows for a single photonic network to be configured as a combination of multiple different power splitters, in which the optical response depends on which one of the input ports receives the optical signal.

We note that with this multi-objective approach, our design framework is also capable of simultaneously optimizing different versions of a given device. This is especially important for the critical capability of building inherent fabrication tolerance in our networks. Specifically, allows for simultaneously optimizing the optical response of multiple different versions of a deep photonic network, each version being the result of a different over-etch or under-etch scenario, thus considering multiple different variations in waveguide widths that may arise from fabrication errors. By integrating this multi-objective approach, our framework can evaluate a combined figure of merit including the device performance not only under ideal conditions, but also with possible fabrication errors.

To represent both input selection-based and fabrication tolerance-related capabilities, we define a more general figure of merit as a mean squared error including all possible combinations of fabrication variations and input ports as

$$J(x) = \frac{1}{Q} \sum_{\Omega} \sum_{\Delta w} \sum_{\lambda} |T_{\text{calculated}}(\Omega, \Delta w, \lambda, x) - T_{\text{target}}(\Omega, \lambda)|^2$$

where the width offset parameter Δw represents the over-etch or under-etch perturbations in waveguide widths, and Ω indicates the input port selection, which now dictates the type of optical operation applied on the input signal. Consequently, the target transfer function $T_{\text{target}}(\Omega, \lambda)$ is now also a function of Ω . For this more general figure of merit, Q is the updated total number of combinations of all wavelengths, etch-offsets, and input port specifications.

This formulation allows us to design networks with more complex relationships between input-output pairs while simultaneously achieving tolerance against fabrication

variations. We showcase this capability by designing a fabrication-tolerant photonic network with two inputs and three outputs, with a combined power splitter functionality, as illustrated by the device schematic in Fig. 6(a). The target functionality for this device is configured such that light entering the center input is separated equally between the three outputs (1/3, 1/3, 1/3), whereas the light entering the top input is separated equally between only the top and bottom output ports (1/2, 0, 1/2) throughout the entire C-band. This network is constructed from four consecutive layers of interferometers as shown, resulting in a total footprint of $8 \times 320 \mu\text{m}^2$.

FIG. 6. Multi-objective optimization of a deep photonic network with multiple different power splitter capabilities and tolerance against fabrication variations. (a) Schematic drawing of the 2-input, 3-output deep photonic network with 4 layers of interferometers. The network acts as a 1×3 splitter for the center input (green arrows), and as a 1×2 splitter for the top input (red arrows). The mean squared error throughout optimization of the network with 144 trainable parameters for (b) the ideal device and (c) the fabrication-tolerant device. Convergence is achieved in less than one minute for both devices. (d), (e) Transmission at the designated output ports of ideal (blue) and fabrication-tolerant (orange) devices as a function of wavelength. Solid lines indicate performance under no fabrication variations, and shaded areas indicate deviation from this performance in case of over-etch and under-etch variations of up to 20 nm. (f) Mean squared error subject to over-etch and under-etch variations for both devices. With ± 20 nm modification of waveguide widths, the resulting error is more than 10x better for the fabrication-tolerant device than the ideal device.

For analysis of fabrication-tolerant design capability, we demonstrate the performance of networks designed both without and with tolerance to fabrication errors. The evolution of figures of merit throughout the optimization processes are plotted in Fig. 6(b) and 6(c). In Fig. 6(c), five different Δw offsets (-20 nm, -10 nm, 0 nm, 10 nm, 20 nm) were considered. In this fabrication-tolerant design, as the optimizer takes into account not a single network but five different networks simultaneously, the resulting figure of merit effectively includes optimizing the transfer function of a total of $5 \times 4 = 20$ MZIs. From this perspective, device optimization under fabrication errors inherently involves scaling to a larger number of interferometers, simply by the nature of this target functionality.

While scaling in such artificial dimensions has obvious practical differences from spatial scaling in network depth or width, the resulting fabrication tolerance capability can be considerably more important for usability in application settings.

For this specific example, the final figures of merit for the ideal and fabrication tolerant networks were 1×10^{-5} and 5×10^{-5} , respectively. As anticipated, the fabrication tolerant device yields slightly worse performance as evidenced by the larger figure of merit. Moreover, increased complexity due to the consideration of multiple objectives for this device results in a greater number of iterations needed for convergence. However, despite doubling the number of iterations, we note that the total optimization time recorded only increases by less than 5 seconds, underscoring the computational efficiency of the design framework (more details can be found in Supplementary Section 4).

More importantly, the performance of the combined power splitter, and a comparison between the ideal and fabrication-tolerant devices is shown in Fig. 6(d) and 6(e). Despite the slightly larger figure of merit for the fabrication-tolerant network, both devices demonstrate near-perfect transmissions under ideal fabrication conditions. However, under nonzero Δw offsets, the fabrication-tolerant device maintains much flatter transmission spectra on all of its output ports throughout the entire C-band. This result demonstrates the ability of deep photonic networks to achieve more complex and multi-functional capabilities while simultaneously enabling much better robustness against fabrication errors across all output ports, for all objectives, through the entire design spectrum. We quantify these built-in fabrication tolerance capabilities further in Fig. 6(f) by plotting the mean squared error as a function of Δw for over-etch and under-etch scenarios ranging from -20 nm to 20 nm. While the ideal device clearly achieves a better absolute error under no fabrication errors ($\Delta w = 0$ nm), the fabrication-tolerant network demonstrates larger tolerances by maintaining a significantly lower figure of merit in the case of nonzero Δw . These results also demonstrate the practicality of our design framework for integration in a wide variety of applications and fabrication platforms, by giving system designers a choice in the final selection between different designs, which can be influenced by the specific fabrication procedures used.

Device Category 2

The physical scalability and the variety/complexity of optical capabilities are further detailed by our demonstration of band-pass filters, which are commonly used in optical communication applications. For these demonstrations we choose target bandwidths of 200 GHz, 100 GHz, and 50 GHz, and demonstrate separate networks for each filter. In order to achieve these metrics that are typically required in communications applications, we expand the deep photonic network structure with longer delays in the

form of spirals as shown in Fig. S7(a). This structure enables band-pass optical transfer functions that are otherwise typically achieved with combinations of ring/disk resonators and/or MZIs [21-26]. The updated simulation of optical response for such a network also includes modular integration of lengths of these spiral waveguides as trainable parameters alongside the existing widths and lengths for the custom tapers. The phase acquired through this waveguide is a function of its trainable length, as illustrated in Fig. S7(b). For this specific application, each network consists of a 1-input 2-output configuration with the labeled through and drop port outputs.

FIG. S7. Design of optical band-pass filters with the deep photonic network architecture. (a) Schematic drawing of the network structure where longer spiral waveguides have been added in each Mach Zehnder interferometer for achieving band-pass functionality. (b) The phase accumulated in the spiral waveguide as a function of wavelength is calculated using its trainable length L_s and the default width of $w_{\text{default}} = 450$ nm. The transmission response at through and drop ports of (c) a 200 GHz band-pass filter with 6 layers of MZIs (150 trainable parameters), (d) a 100 GHz band-pass filter with 9 layers of MZIs (225 trainable parameters), and (e) a 50 GHz band-pass filter with 10 layers of MZIs (250 trainable parameters including trainable spiral waveguide lengths). The simulated final 3dB bandwidths are 201.90 GHz, 96.09 GHz, and 50.51 GHz for the three devices, respectively. Optimization for each one of the filters converges in several hundred iterations, in under three minutes of total computation time. All devices are optimized between 1547 nm and 1553 nm.

Using this structure, we have designed three separate optical band-pass filters at a center wavelength of 1550 nm, with target bandwidths of 200 GHz, 100 GHz, and 50 GHz, whose simulated responses are shown in Fig. S7(c)-(e), respectively. The target bandwidth for each filter was controlled by specifying a range of wavelengths at which the entire optical input was transmitted to the drop port around the center wavelength. Inherently, achieving narrower filter bandwidths while maintaining a good extinction ratio represents a more difficult problem, and therefore, requires greater degrees of optical design freedom. Consequently, 6-layer, 9-layer, and 10-layer networks were used for designing the three filters, in decreasing order of optical bandwidth. The demonstrated bandwidths for the optimized devices were 201.90 GHz, 96.09 GHz, and 50.51 GHz for the three filters shown. At the center wavelength of filter transmission, extinction ratios of -48 dB, -25 dB, and -35 dB were obtained. As indicated by these transmission

results, all three filters demonstrate agreement with their target specifications and achieve sufficiently high extinction ratios for communication applications. In contrast to our power splitter examples, we have included a dB-based specification of optical transmission between the through and drop ports in these band-pass filter demonstrations. From an implementation perspective, this allows the optimizer to target a given maximum cross-talk between the outputs, and modify trainable width and length parameters towards achieving this goal. With this specification, our filters have achieved better than -20 dB cross-talk between the two outputs for the majority of design spectrum.

Before optimization, spiral waveguide lengths were randomly initialized using $L_s = L_{\text{default}} - p_L \cdot \Delta L$, where p_L is a random variable between 0 and 1, and ΔL is a user chosen maximum deviation amplitude from the default, similar to the initialization of the lengths and widths of the custom tapers. After a set of optimization trials with different initializations, as is commonplace in machine learning model training [5-7], in order to achieve gradually narrower filter bandwidths, L_{default} of 50 μm , 80 μm , and 155 μm were used, respectively for the three devices. As anticipated, longer spiral lengths yielded photonic networks that are more suitable as narrow-bandwidth filters in our optimizations. However, the exact dependence of filter bandwidth on waveguide lengths remains a more complex function of optical power transmitted through each arm, which is influenced by the phase relationships between all custom tapers and spirals. The maximum final optimized spiral waveguide lengths remained below 55 μm for the 200 GHz filter, and 160 μm for all other filters designed. Due to the compact structure of the spiral geometry, their placement can be planned in rectangular blocks adjacent to one of the interferometer arms as shown, with only a minor impact on the overall device footprint.

These examples demonstrate the capability of our framework to efficiently design optical band-pass filters with varying bandwidths and high extinction ratios, further expanding the scalability, practicality, and applicability of our approach through more complex functions particularly in the context of optical communication systems.

Device Category 3

As Reviewer #2 suggested, the demonstration of specific phase/dispersion profiles is another area where our photonic networks can excel and provide unique capabilities. To address this point, we designed photonic networks providing constant dispersion profiles with the details given below.

The goal of these photonic networks is to provide a linearly changing group delay, in order to achieve constant dispersion. Traditionally, on-chip constant dispersion is achieved using components like cascaded ring resonators [27-29] or Bragg gratings

[30-32] due to highly linear group delay within their transmission/reflection bandwidths. At the wavelengths within their acceptable dispersion compensating bandwidth, these devices can inherently achieve near unity transmission, by the nature of their operation principles. A similar functionality can be accomplished by deep photonic networks; but this requires formulation of a combined objective with separately specified transmission and dispersion targets, in a fashion similar to the multi-objective designs we demonstrated. To do so, we separate the complex output of a photonic network into its amplitude and phase components, enabling calculation of transmission and group delay (as well as dispersion) results at each output port. For calculation of dispersion, the first and second derivatives of phase recorded and unwrapped at respective output ports are computed, using twice differentiable interpolations of effective index and directional coupler response. Compared to our transmission-only objectives, here we use a narrower wavelength spacing of 0.1 nm in order to avoid potential undersampling issues during unwrapping of this phase. We then define a figure of merit including all of these metrics to quantify the difference between target and calculated transmission and dispersion responses as

$$J(x) = \frac{1}{Q} \sum_{\lambda} \left((1 - \eta) |T_{\text{calculated}}(\lambda, x) - T_{\text{target}}(\lambda)|^2 + \eta |D_{\text{calculated}}(\lambda, x) - D_{\text{target}}(\lambda)|^2 \right)$$

where T and D indicate the transmission and dispersion responses of the network as a function of wavelength λ and trainable parameters x , and we defined an additional scaling parameter η . The control of this parameter η allows the designer to specify a relative weight between the transmission and dispersion objectives. We find experimenting with this parameter to be particularly useful in order to optimize for a certain amount of dispersion within the desired bandwidth, while still minimizing the power lost to other output ports to achieve low insertion loss (near unity transmission). For optimizing devices with this given figure of merit, we initialize and iteratively update trainable lengths and widths of a deep photonic network as before.

As proof of principle, we demonstrate two networks with a target constant dispersion of 0.5 ps/nm, with bandwidths of 3 nm and 6 nm, centered at 1550 nm, using 7 layers and 9 layers of MZIs, respectively. The geometry of these dispersion-compensating networks is structurally identical to the band-pass filters in Fig. S7(a); but the drop port remains unused as the entire output is collected from the through port in these devices. Here, the spiral waveguides were initialized to be around 47 μm in length, and then were iteratively optimized together with the custom tapers in the photonic network. The resulting group delays are plotted in Fig. S8(a) and S8(d), with both devices illustrating linear delay profiles confirming successful device optimizations. The dispersion is calculated from the derivative of this group delay, and plotted for the two devices in Fig. S8(b) and S8(e). The dispersion profiles deviate from the constant target of 0.5 ps/nm by 35-40 fs/nm towards the end of the specified bandwidth. Overall, the narrower-band device achieves better agreement with the dispersion target specified, due to the relative difficulty of achieving constant dispersion across wider optical bandwidths.

A similar observation can be made regarding the transmission results in Fig. S8(c) and S8(f). The 9-layer device with 6 nm of target constant dispersion spectrum experiences slightly higher insertion loss, especially towards the end of the specified spectral range. As before, this emphasizes the relative difficulty of the target specification as simultaneously satisfying transmission and dispersion objectives across a wider bandwidth represents a more difficult optimization problem. Still, we observe better than -0.4 dB of insertion loss throughout the majority of the operation bandwidth, which is consistent with previous results from the literature [30, 33-35].

For compensating higher amounts of dispersion, multiple copies of the same networks can be cascaded in series, as their insertion loss remains relatively small. In comparison, while the demonstrated amounts of dispersion compensation in literature can exceed 10 ps/nm [30, 34-36], our network structure allows for not only constant anomalous dispersion for dispersion compensating applications, but also specification of arbitrary target dispersion profiles. These capabilities allow for unique design freedom in pulse-shaping [37-39] and quantum information processing applications [40], which typical dispersion compensators cannot provide. Even though cascaded ring resonators technically have the capability to achieve arbitrary dispersion profiles [27], their limitations become apparent when considering that each ring typically offers just two design parameters, thereby severely restricting the degrees of design freedom compared to our architecture.

7-Layer Deep Photonic Network

9-Layer Deep Photonic Network

FIG. S8. Examples of deep photonic networks for dispersion optimization. Simulated group delay as a function of wavelength for (a) a 7-layer deep photonic network with 175 trainable parameters, and (d) a 9-layer deep photonic network with 225 trainable parameters (these parameters include the lengths and widths in custom tapers as well as the lengths of spiral waveguides). The network structure is identical to the one illustrated in Fig. S7(a); but the entire output is collected from the

through port. **(b), (e)** Spectral dispersion profiles of the two optimized devices. Both devices demonstrate near-constant dispersion throughout their optimized bandwidth of 3 nm and 6 nm, respectively. A target constant dispersion of 0.5 ps/nm was used. **(c), (f)** Transmission through the devices remains better than -0.4 dB for the majority of their optimized spectrum. Some more insertion loss is observed towards the end of the specified spectral range. (GDD: group delay dispersion)

In summary, these three categories of additional devices we included, and further details we provided in Fig. S4 on the performance of larger photonic networks collectively demonstrate multiple dimensions of scalability and applicability for achieving more complex photonic functions.

Action taken:

We have included these new results in multiple different sections throughout the main text and our supplementary materials:

- The 1×8 power splitter with arbitrarily-selected transmission targets in Fig. S4, and corresponding explanations have been added in Supplementary Section IV.
- A new section titled “Multi-Objective Design Capabilities” and the new Fig. 6 were added with the details of our combined power splitter device explained above.
- The band-pass filter results and the new Fig. S7 were added in the new Supplementary Section VI.
- We have added the details of the dispersion optimization and demonstrated devices in Fig. S8 in the new Supplementary Section VII.

Reviewer #2:

“2) Comparison with the literature must be made in order to claim that experimental results are state of the art. This should also include footprint as a criterium. Also, to show that deviations are "minimal" some references should be given. For instance, the transition in the duplexer spectral function is not very sharp, therefore spectral deviation may appear minimal.”

Our Response:

The reviewer correctly points out that device footprint is an important consideration, in addition to the comparison of experimental performance shown. Similar suggestions were also made by the other reviewers too. To address these, we first start by summarizing the performance of our experimentally measured devices, together with comparable devices from literature, and point out specific performance metrics of our networks representing state-of-the-art. Then, we provide some perspectives regarding the capability vs. footprint tradeoff for existing device structures and techniques in the literature and also for our demonstrated design framework.

Our 50/50 and 75/25 power splitters demonstrate simulated 1dB bandwidths of over 200 nm, and experimentally measured 1dB bandwidths as wide as the entire measured spectrum of 120 nm. Both devices operate with insertion losses below 0.61 dB. In

comparison to previous experimental demonstrations [40-50], these metrics represent the state-of-the-art performance in bandwidth, and illustrate comparable performance in insertion loss. Likewise, our duplexer demonstrates better experimental performance than devices with similar functionality [9, 14, 51, 52], with less than 0.66 dB insertion loss, flat-top transmissions at both outputs, and a cutoff wavelength shift of only 5 nm. Despite the operation bandwidth reaching over 120 nm, this achieved spectral shift is also similar to reported metrics from literature where specific cutoff wavelengths for resonators, filters, or duplexers typically deviate from their targets by several nm [9, 14, 51-54]. Depending on specific application requirements, this shift can be compensated through standard thermal tuning mechanisms [55, 56]. Similarly, based on application needs, the roll-off between the two bands may also be improved by optimizing with a tighter spectral placement of transmission targets shown in Fig. 3(i).

These previously reported splitters and duplexers range approximately between $5 \mu\text{m}^2$ and 6mm^2 in footprint, depending on their operation principles and constituent waveguide structures. While some of these previous demonstrations using basic ring resonators [53, 54, 57-59], Y-junctions [44, 50, 60, 61], and subwavelength grating waveguides [43, 45, 49, 62] can achieve functionally similar operation within smaller footprints than our deep photonic networks, their capabilities remain limited to well-defined and fundamental operations with potentially narrower operation bandwidths. Even for devices obtained with free-form inverse-design techniques [9, 13, 14, 40, 63, 64], the types, complexity, and bandwidth of possible optical operations are practically restricted by the inherent computational difficulty of addressing complicated objectives that require greater degrees of freedom and larger device sizes. In contrast, our networks naturally scale to a greater number of input-output pairs, with little change in their computational optimization performance (Supplementary Section 4). As a result, deep photonic networks allow for a wide and diverse array of demonstrated functional capabilities as complex as arbitrary, multi-functional, and inherently fabrication-tolerant power splitters, duplexers, band-pass filters, and dispersion compensators. As such, these networks not only advance the state-of-the-art in device performance, but also create new pathways for custom photonic system solutions.

Action taken:

The above explanations have been added in our discussion.

Reviewer #2:

“When quoting “excess losses”, do authors refer to on-chip losses, thereby removing coupling losses? How do they calibrate the measurement? In fig. 4a,b, there is a

structure of oscillations which might be due to Fabry Perot interference due to residual reflection (at the fiber couplers). I would expect some comments here.”

Our Response:

We thank the reviewer for pointing out that more details regarding the experimental measurement procedures are necessary. To avoid any confusion, we have eliminated mixed use of “excess insertion loss” and “excess loss” from the manuscript, and replaced these with “insertion loss”. Regarding the characterization of this loss, we have provided relevant details below, and also added them in a new Methods section in the manuscript.

For experimental characterization of deep photonic networks, our measurement procedures include standard steps to remove any losses due to on- and off-chip coupling of optical signals through grating couplers such as reflections [71] or potential mismatches between fiber, grating, or waveguide modes [72]. Our reported insertion losses refer to only the additional losses through the photonic networks themselves, after these grating coupler losses have been removed. The coupling losses have been measured at four separate fiber zenith angles between 8° and 14° , using grating coupler test structures on the same chip as the measured deep photonic networks, and then combined together in order to accurately characterize as wide a measurement bandwidth as possible. All measurements have been performed using a continuous-wave tunable laser source (Santec TSL-710), an optical power meter (Santec MPM-210), and a polarization controller. The tunable source was operated using a wavelength sweep from 1480 nm to 1600 nm with a sampling rate of 40 ps to obtain the transmission characteristics of the measured structures.

The spectral oscillations in our experimental measurements indicate presence of well known Fabry-Perot interference due to reflections at fiber-to-chip interfaces [73]. These reflections are an inherent result of characterizing the devices on their own, with grating couplers directly connected to the inputs and outputs of our deep photonic networks. As parts of a larger photonic system, the networks can be directly connected by waveguides to other upstream and downstream on-chip devices, eliminating potential reflections at the grating interfaces and any associated spectral oscillations.

Action taken:

We have added the two paragraphs above in a new Methods section titled “Experimental Measurements”.

Reviewer #2:

“3) I think that the description of the mathematical model should be improved. For instance eq. 2 describes a chain of MZI (as in fig 3) but not the more general mesh described in fig 1.”

Our Response:

The reviewer is correctly referring to how the product of transfer matrices given in Eq (2) does not technically generalize to larger networks. Before we detail and improve this mathematical model, we note that even for larger networks, it has recently become common practice to express the appropriately ordered and arranged combination of transfer functions using just a single product as we have in Eq (2) [29-31]. Here, instead of writing the products separately, the variable Γ is used to indicate the placement and ordering of specific transfer matrices (T_{ij}) according to the network topology.

Nevertheless, it is straightforward to generalize this model to networks with a greater number of inputs and outputs. For larger mesh topologies such as those shown in Fig. 1 (and also in Fig S6), the overall scattering matrix $S(\lambda)$ is given by

$$\prod_{q=1}^{\lfloor \frac{M}{2} \rfloor} \begin{bmatrix} T_{1,R+1-2q} & & & \\ & \ddots & & \\ & & & T_{n,R+1-2q} \end{bmatrix} \begin{bmatrix} F & & & \\ & T_{1,R-2q} & & \\ & & \ddots & \\ & & & T_{n-1,R-2q} \\ & & & & F \end{bmatrix}$$

for networks with an even number of inputs, and by

$$\prod_{q=1}^{\lfloor \frac{M}{2} \rfloor} \begin{bmatrix} F & & & \\ & T_{1,R+1-2q} & & \\ & & \ddots & \\ & & & T_{n,R+1-2q} \end{bmatrix} \begin{bmatrix} T_{1,R-2q} & & & \\ & \ddots & & \\ & & & T_{n,R-2q} \\ & & & & F \end{bmatrix}$$

for networks with an odd number of inputs. Here, M represents the number of interferometric layers, $n = \lfloor N/2 \rfloor$, $R = 2\lfloor M/2 \rfloor + 1$ and F is a scalar indicating the phase accumulated through the topmost and bottommost arms of the network where no interferometer is present. Note that the very first term is omitted from the products when using an odd number of layers.

Action taken:

In the revised manuscript, Eq. 2 has been replaced with the two expressions given above. The explanations for the relevant terms have also been updated accordingly after these expressions, ensuring generalizability of the mathematical model to larger network sizes.

Reviewer #2:

As a minor point, are authors sure that "scattering matrix" is appropriate here? Also, neglecting reflection, although plausible and desired, need some justification. The important assumption is that the arbitrary profiles do not induce any reflection, which might deserve some comment. Besides, did authors measure the reflected light in their circuits?"

Our Response:

The reviewer points out that typical usage of the "scattering matrix" terminology generally includes transmission and reflection components. In our analysis, the scattering formulation only includes the linear transformations indicated by T_{ij} between the input and output pairs, where no input-to-input "scattering" in the form of reflections has been considered. Below, we show the validity of this assumption by providing both simulation and experimental results corroborating our findings.

We start by examining reflection in our interferometers through 3D-FDTD simulations shown in Fig. S5 from our 50/50 power splitter, and one of its optimized tapers. In simulation, the entire network's response in Fig. S5(a) indicates a total reflection of less than -50 dB throughout its entire bandwidth. In Fig. S5(b), we plot the reflection from just a single optimized taper (as shown in Fig. 1(d)), but make sure to select the taper with the widest width variations along its length to observe the worst possible reflections. For this single taper, the simulated reflections remain below -60 dB, indicating no practical influence of input-to-input "scattering" effects on the performance of demonstrated deep photonic networks. It is important to note that the width regularizers explained in Supplementary Section 2 play an important role in limiting the deviation of widths in a given taper, which also effectively minimize reflections and other scattering losses.

FIG. S5. 3D-FDTD simulation results of reflection in deep photonic networks. (a) Simulated reflection measured at the input of the 50/50 power splitter network. **(b)** Simulated reflection for one of the optimized tapers in the same photonic network.

In addition to the simulations above, we also conducted experimental measurements of reflection, using the same 50/50 splitter network as an example. As the reviewer asked, we first measure the reflection from our deep photonic network (dpn) with input and output grating couplers (gc) using a fiber circulator, yielding the combined reflection spectrum of the “gc+dpn+gc” structure. From this measurement, we extract the reflection spectrum corresponding only to the photonic network, through the following back-calculation procedure involving the measured reflection and transmission spectra of the “gc+gc” grating coupler input-output test structure on the same chip.

The reflection-included scattering matrix for a grating coupler can be expressed as

$$S_{gc+gc} = \begin{bmatrix} t_{gc} & -jr_{gc} \\ -jr_{gc} & t_{gc} \end{bmatrix}$$

where t_{gc} and r_{gc} are the transmission and reflection amplitude coefficients of the grating coupler, respectively. The combined scattering matrix for the “gc+gc” grating coupler test structure is calculated by the product of the corresponding transfer matrices, and is given by

$$S_{gc+gc} = TS(ST(S_{gc}) \times ST(S_{gc})) = \frac{1}{1 + r_{gc}^2} \begin{bmatrix} t_{gc}^2 & -jr_{gc}(1 + r_{gc} + r_{gc}^2) \\ -jr_{gc}(1 + r_{gc} + r_{gc}^2) & t_{gc}^2 \end{bmatrix}$$

where we defined $ST()$ and $TS()$ functions to convert between scattering and transfer matrices for convenience [20]. Using this description, and the experimental results of transmission and reflection from our grating coupler test structure, we extract t_{gc} and r_{dpn} parameters as a function of wavelength.

Through a similar procedure, we then express the scattering matrix for the measured “gc+dpn+gc” device as

$$S_{gc+dpn+gc} = TS(ST(S_{gc}) \times ST(S_{dpn}) \times ST(S_{gc}))$$

where

$$S_{dpn} = \begin{bmatrix} t_{dpn} & -jr_{dpn} \\ -jr_{dpn} & t_{dpn} \end{bmatrix}$$

represents the reflection-included scattering matrix for the input and one of the outputs of the deep photonic network, with corresponding transmission (t_{dpn}) and reflection (r_{dpn}) amplitude coefficients.

This expression yields

$$S_{gc+dpn+gc} = \frac{1}{1 + 2r_{gc}r_{dpn} + r_{gc}^2(r_{dpn}^2 + t_{dpn}^2)} \begin{bmatrix} A & B \\ B & A \end{bmatrix}$$

where

$$A = t_{gc}^2 t_{dpn}$$

and

$$B = -j \left[2r_{gc}^2 r_{dpn} + r_{dpn} t_{gc}^2 + r_{gc}^3 (r_{dpn}^2 + t_{dpn}^2) + r_{gc} (1 + r_{dpn}^2 t_{gc}^2 + t_{dpn}^2 t_{gc}^2) \right]$$

which is then used to back-calculate t_{dpn} and r_{dpn} parameters as a function of wavelength, using the previously extracted t_{gc} and r_{gc} from above. The results are plotted in Fig. S6 showing the experimentally obtained reflection and transmission spectra from the 50/50 splitter deep photonic network. The measured reflection from the entire deep photonic network remains below a maximum of -16.6 dB, and around -30 dB for the majority of the spectral range as shown in Fig. S6(a). Similarly, the transmission result now recalculated by taking reflections into account in Fig. S6(b) closely resembles the directly-measured transmission in Fig. 4(a), as also replotted in Fig. S6(b) for reference. These results verify practically negligible impacts of reflection from our deep photonic networks and their constituent components on the network performance.

FIG. S6. Experimentally measured reflection and transmission results from the 50/50 splitter deep photonic network. (a) Reflection spectrum measured at the input of the network. **(b)** Transmission spectra at one of the outputs of the network, calculated by taking potential non-zero reflections into account (blue), and its comparison with the direct measurement assuming no reflections (gray).

Action Taken:

These new simulation and experimental results together with the accompanying mathematical description have now been added in a new Supplementary Section V, titled “Characterization of Reflection in Deep Photonic Networks”.

Reviewer #2:

“Overall, the manuscript is well prepared, the quality of the figures is high and the methodology is sound. I suggest authors improve the introduction which I found not very focused and not particularly useful to describe the scope and the purpose of this work. It could be more succinct in my perspective.

Our Response:

We appreciate Reviewer #2 for acknowledging the efforts we put into the organization of our figures and the manuscript. Regarding the introduction, both Reviewer #2 and Reviewer #3 have suggested reducing the amount of introductory material we presented. Upon their suggestions, we have shortened our introduction and included a more focused set of prior work to motivate the research direction we presented.

Action Taken:

Basic device design specifics and algorithmic details have now been removed from our introduction. The first three paragraphs of the introduction now only include several example capabilities of inverse-designed structures and MZI networks. Here, we briefly motivate the need for arbitrarily complex, ultra-broadband/wavelength-specific devices, and mention critical computational requirements. We then finish this section by introducing deep photonic networks, and summarizing our findings as before.

Reviewer #2:

Were authors to demonstrate an example where complexity and scalability are apparent, with respect for instance to what is already available (for instance the reprogrammable photonic circuits implementing an unitary matrix transform with a fairly large number of channels), then they would make a strong case for publication. At this stage I am not yet convinced the results are up to the level of Nature Communications.”

Our Response:

We would like to thank Reviewer #2 for highlighting the importance of complexity and scalability once again through their constructive comments. As we explained in our detailed responses above, to specifically target these suggestions in our updated submission, we have now included multiple new categories of devices:

- Multi-Objective Functionality: Our demonstrations now include a combined 1×2 and 1×3 splitter with simultaneous tolerance capabilities for fabrication errors (Fig. 6).
- Band-Pass Filters with Different Bandwidths: We have further demonstrated how deep photonic networks can be used to create band-pass filters of example bandwidths ranging from 200GHz to 50 GHz (Fig. S7).
- Devices with Optimized Dispersion Profiles: We added a set of “all-pass” networks enabling target objectives of arbitrary dispersion spectra, and demonstrated constant dispersion profiles (Fig. S8).
- Extended Network Architectures: We have illustrated the transmission response of larger photonic networks extending up to 10 layers and 8 output ports, optimized for arbitrary power splitting applications (Fig. S4).

These new devices demonstrate a much more versatile and complex array of optical functionalities, with scalability to greater device sizes and input-output pairs as well as multi-objective optimization capabilities. We believe these added demonstrations now fully address the capability and scalability aspects mentioned by Reviewer #2, and illustrate applicability of deep photonic networks for a wide range of applications and systems in optical communications, sensing, and computing.

—————Reviewer #3—————

Reviewer #3:

“Optimization techniques are now the standard in photonic component design. Previous work in this area includes [1-3]. There are multiple works that specifically use backpropagation for photonic device design.

The authors used optimization techniques to aid in the design of three integrated photonic components (with specific spectral responses) that were fabricated. In addition, they developed a simulation framework for chaining together these components and creating “meta” devices. The optimizer, which leverages backpropagation, is controlling “trainable widths” that correspond to the physical geometry of the device. Overall, the authors have produced a design/simulation framework for multi-port linear optical photonic systems that optimizes over device parameters.”

Our Response:

We thank Reviewer #3 for reading our manuscript in detail and providing constructive comments. As they mentioned, the availability and popularity of machine learning techniques and the ability to access necessary hardware resources for these

optimization methods continue to drive innovation in the design of many physical systems, including integrated photonic components. In fact, the use of backpropagation (or automatic differentiation in general), and the resulting computational efficiency are key enabling factors for rapid simulation/optimization of photonic device response, and also for our subsequent demonstrations of devices with a wide array of functionalities.

Reviewer #3:

“The paper is well written and understandable due to its clear organization. However, the amount of introductory material is unnecessary.”

Our Response:

Similar to Reviewer #2’s comment above, we also sincerely appreciate Reviewer #3 for acknowledging the efforts we put in the organization of our manuscript. Both reviewers have suggested reducing the amount of introductory material we presented. Upon their suggestions, we have shortened our introduction and included a more focused set of prior work to motivate the research direction we presented.

Action Taken:

Basic device design specifics and algorithmic details have now been removed from our introduction. The first three paragraphs of the introduction now only include several example capabilities of inverse-designed structures and MZI networks. Here, we briefly motivate the need for arbitrarily complex, ultra-broadband/wavelength-specific devices, and mention critical computational requirements. We then finish this section by introducing deep photonic networks, and summarizing our findings as before.

Reviewer #3:

“The components produced by this method of chaining devices together into “deep photonic networks” did not produce record-breaking results. This is in contrast to the work shown in [2] and I believe publication of this manuscript in Nature Communications would require not only a new method, but a record breaking device/subsystem that resulted from its use.”

Our Response:

Reviewer #3 points out that similar functionality in power splitting or spectral splitting has been achieved with other types of devices reported in literature. Naturally, as these splitting operations serve as common benchmarks for evaluating different photonic design approaches, many devices have been previously used for achieving similar optical functionality. However, our experimental demonstrations do in fact exhibit

best-in-class performance metrics for their respective photonic operations. Namely, our 50/50 and 75/25 power splitters demonstrate simulated 1dB bandwidths of over 200 nm, and experimentally measured 1dB bandwidths as wide as the entire measured spectrum of 120 nm. Both devices operate with insertion losses below 0.61 dB. In comparison to previous experimental demonstrations [40-50], these metrics represent the state-of-the-art performance in bandwidth, and illustrate comparable performance in insertion loss. Likewise, our duplexer demonstrates better experimental performance than devices with similar functionality [9, 14, 51, 52], with less than 0.66 dB insertion loss, flat-top transmissions at both outputs, and a cutoff wavelength shift of only 5 nm. Despite the operation bandwidth reaching over 120 nm, this achieved spectral shift is also similar to reported metrics from literature where specific cutoff wavelengths for resonators, filters, or duplexers typically deviate from their targets by several nm [9, 14, 51-54]. Depending on specific application requirements, this shift can be compensated through standard thermal tuning mechanisms [55, 56]. Similarly, based on application needs, the roll-off between the two bands may also be improved by optimizing with a tighter spectral placement of transmission targets shown in Fig. 3(i).

These previously reported splitters and duplexers range approximately between 5 μm^2 and 6 mm^2 in footprint, depending on their operation principles and constituent waveguide structures. While some of these previous demonstrations using basic ring resonators [53, 54, 57-59], Y-junctions [44, 50, 60, 61], and subwavelength grating waveguides [43, 45, 49, 62] can achieve functionally similar operation within smaller footprints than our deep photonic networks, their capabilities remain limited to well-defined and fundamental operations with potentially narrower operation bandwidths. Even for devices obtained with free-form inverse-design techniques [9, 13, 14, 40, 63, 64], the types, complexity, and bandwidth of possible optical operations are practically restricted by the inherent computational difficulty of addressing complicated objectives that require greater degrees of freedom and larger device sizes. In contrast, our networks naturally scale to a greater number of input-output pairs, with little change in their computational optimization performance (Supplementary Section 4). As a result, deep photonic networks allow for a wide and diverse array of demonstrated functional capabilities as complex as arbitrary, multi-functional, and inherently fabrication-tolerant power splitters, duplexers, band-pass filters, and dispersion compensators.

From this perspective, our deep photonic network infrastructure provides a tractable path towards realistically achieving arbitrary linear optical functionality for any application, which is currently not possible with any given approach, to the best of our knowledge. We have illustrated the diversity of these capabilities with the newly added categories of devices we included in our revised manuscript. In addition to achieving state-of-the-art performance metrics, the ability of the presented framework to streamline design workflows, simultaneously enable inherent fabrication tolerance for multi-objective devices, and create a tractable path towards customizable, large-scale photonic system design can even be more important practical considerations for many industrial applications. As such, we believe that deep photonic networks not only

advance the state-of-the-art in device performance, but also create new pathways for custom photonic system solutions, making them a pivotal advancement in the field of integrated optics.

Action Taken:

The first two paragraphs from our response above have now been added in our discussion, with detailed comparisons of photonic network performance with comparable demonstrations from literature.

Reviewer #3:

“The components demonstrated here are impractically large for industrial applications and would suffer significantly from phase disorder introduced by side wall roughness (requiring phase tuning post fabrication to yield the correct transfer matrix).”

Our Response:

Reviewer #3 correctly points out that potential phase disorder resulting from sidewall roughness is an important and well known challenge to overcome, especially in photonic platforms with high index contrast. In this regard, we first note that in our experimental demonstrations using the 220 nm-thick commercially available SOI platform from IMEC [49], we have observed no detrimental effects on device performance due to this phenomenon for our networks in Fig. 4. Up to a network depth of 6 layers, our measured experimental performance has shown good agreement with expected optimization results; and our observed deviations have remained consistent with the fabrication tolerance analyses we have performed (as also indicated by Reviewer #2).

From a more general perspective, as specifically highlighted by Reviewer #3 and also suggested by the first two reviewers, several challenges related to network footprint exist such as propagation losses, accumulation of fabrication errors, and sidewall roughness-induced phase disorder. These challenges have important implications for use of our devices in practical, industrial applications. We address these considerations below, and provide perspectives for each specific factor we listed.

- Of these factors, we expect typical SOI waveguide propagation losses of 2-3dB/cm not to practically introduce any additional design considerations specific to our photonic networks, since any propagation-related losses remain balanced between our symmetrically-placed interferometer arms by design. Using an MZI length of 80 μm , even for the longest, 10-layer device we have shown in Fig. S4, this corresponds to an expected total loss of approximately

$3\text{dB/cm} \times 80 \mu\text{m} \times 10 = 0.24 \text{ dB}$, which is acceptable for many on-chip optical applications. It is also important to note that the symmetric design of all interferometer arms in a given layer ensures propagation through an equal number of directional couplers and S-bends for each arm. The expected total loss through these individual components is already extracted from 3D-FDTD simulations, and is included in our analysis shown in Fig. 5 and Fig. S2. Up to a network depth of 10 layers, these plots illustrate that the optimized figures of merit remain below 10^{-4} (in line with the figures of merit obtained for the fabricated devices), indicating no practical influence of the added scattering losses due to the S-bends or directional couplers.

- Accumulation of fabrication errors through cascaded nodes presents an important challenge for any mesh-like structure. For deep photonic networks, since each individual device simulation can be performed on the order of milliseconds, our design framework enables building inherent tolerance to fabrication errors, as already shown in Fig. 6. Through this approach, the custom tapers are automatically optimized by taking into account multiple different under-etched and over-etched versions of the same device. We expect this added capability to provide significant flexibility in selecting the most appropriate designs for a specific task, which may also be influenced by fabrication methods used. However, as we have already discussed, longer devices naturally exhibit worse tolerances to fabrication errors accumulating through multiple interferometric layers (see Fig. 5(b), Fig. S2(b), and Fig. S2(d)). Therefore, for networks deeper than 10 layers, we expect practical limitations towards achieving similar levels of fabrication tolerance, which may require electrical, post-fabrication tuning.
- Longer devices also suffer from potential phase errors due to sidewall roughness in the fabricated waveguides. This is an especially well-known problem in high-index contrast platforms, as the larger index difference results in phase errors greater in magnitude. This is also an important reason behind the preference of certain optical phased array demonstrations toward SiN guiding platforms instead of Si, due to the comparatively lower index contrast and the smaller magnitude of resulting phase errors [45-48]. Using the 220 nm-thick commercially available SOI platform from IMEC [49], we have observed no detrimental effects on device performance due to this phenomenon for our experimentally demonstrated networks in Fig. 4, up to a network depth of 6 layers. In case of phase errors for deeper networks, a similar approach to implementing optical phase arrays in SiN platforms can be undertaken, since our design framework remains generalizable to other waveguide materials or

platforms. While the diversity of optical operations and capabilities in design freedom we have shown can provide solutions for many optical applications within less than 10 layers, the presented design framework can be easily generalized to other guiding materials with smaller phase errors. It may also be possible to implement inherent fabrication tolerance to target these phase errors, in a similar fashion to achieving near-optimal optical responses for over-etch/under-etch scenarios discussed above.

Action Taken:

In order to address the phase disorder issue and more general considerations regarding device footprint, we have added the details above in a new Supplementary Section VIII, titled “Comparison and Limitations of Functional Network Capability”.

Reviewer #3:

Overall, I believe this is solid, scientific work but it does not represent a breakthrough that would justify publication in this journal.

Our Response:

We would like to thank Reviewer #3 once again for taking the time to evaluate our manuscript in detail. As we explained in our responses above, our power and spectral splitters demonstrate best-in-class device performance metrics, even though similar demonstrations in literature naturally exist for these benchmark optical functionalities. In addition to these, to better illustrate the diversity of optical capabilities enabled, we have now included multiple new categories of devices in our updated manuscript:

- Multi-Objective Functionality: Our demonstrations now include a combined 1×2 and 1×3 splitter with simultaneous tolerance capabilities for fabrication errors (Fig. 6).
- Band-Pass Filters with Different Bandwidths: We have further demonstrated how deep photonic networks can be used to create band-pass filters of example bandwidths ranging from 200GHz to 50 GHz (Fig. S7).
- Devices with Optimized Dispersion Profiles: We added a set of “all-pass” networks enabling target objectives of arbitrary dispersion spectra, and demonstrated constant dispersion profiles (Fig. S8).

- Extended Network Architectures: We have illustrated the transmission response of larger photonic networks extending up to 10 layers and 8 output ports, optimized for arbitrary power splitting applications (Fig. S4).

These new devices showcase a diverse and complex range of optical functionalities, offering scalability to larger device sizes and greater input-output pairs, along with the capability of multi-objective and fabrication-tolerant optimization. As a result, we believe that deep photonic networks not only advance the state-of-the-art in device performance, but also pave the way towards the transformational prospect of practical and customizable large-scale photonic system design, with substantial implications for a wide range of industrial applications.

REVIEWERS' COMMENTS

Reviewer #1 (Remarks to the Author):

Dear authors, thanks for all the changes and explanations now included in the text. I believe that the pair is now suitable for publication.

Reviewer #2 (Remarks to the Author):

In their revised report, authors address the criticisms by the referees by providing new results which make in my opinion a case for publication.

Essentially, the referees asked for more convincing results to support the claim of scalability and complexity. I think this time authors have addressed these points and they provide a tool which is likely to help the community in designing better photonic circuits.

I therefore recommend publication

Reviewer #3 (Remarks to the Author):

While I appreciate the significant effort that has gone in to responding to the reviewer comments, I fundamentally do not believe the work contributes in a significant way to the body of knowledge in the space. This work does not warrant publication in Nature Communications.